# Endothelial-erythrocyte glycocalyx exchange enables liquid biopsies of endothelial function

Matthew J. Butler [1] ✉, Raina R. Ramnath [1], Michael Crompton [1], Jasmine Aldam[1], Monica Gamez [1], Colin Down[1], Charley Heffer [1], Chris Neal[1], Jialu Li[1], Yan Qiu[1], Laura Carey [1], Laura Skinner[1], Stephen Cross [2], Yamaguchi Yu [3], Judit Sutak[4], Victoria Bills[5], Gavin I. Welsh [1], Rebecca R. Foster[1] & Simon C. Satchell [1]

The luminal surface of blood vessels is covered by a hydrated mesh of sugars and proteins termed the endothelial glycocalyx. The glycocalyx forms a permeability barrier and helps regulate leucocyte migration. Directly detecting glycocalyx damage could provide a major advance in vascular health monitoring. Here we show that red blood cell glycocalyx mirrors the endothelial glycocalyx in health and disease. Using peripherally sampled blood, we confirm that red blood cell glycocalyx measurements predict cardiac and renal endothelial glycocalyx alterations and direct measures of endothelial barrier function in male rats. To investigate the underlying mechanism, we use Azide-Alkyne cycloaddition ('Click' chemistry) to confirm that contact between endothelial and red blood cells results in continual reciprocal transfer of glycocalyx components. These discoveries facilitate real-time monitoring of endothelial damage in patients whilst simultaneously providing a potential explanation as to how red blood cells maintain their glycocalyx during circulation.

Endothelial damage contributes to multiple human diseases[1-3]. One of the earliest manifestations of endothelial damage is loss of the delicate sugar-dense endothelial glycocalyx (eGlx)[2,4-6]. The structure of the eGlx varies between locations, vessel types and organs, according to the specialised functions required[7]. However, 'core' elements are universally expressed[8]. The eGlx contributes to endothelial barrier function, regulates leucocyte binding and migration and 'senses' shear stress[9]. In the kidney glomerulus, eGlx damage increases albuminuria—a hallmark of kidney disease and a strong predictor of cardiovascular mortality[10]. eGlx damage has also been implicated in the pathogenesis of sepsis[11], preeclampsia[12,13], thrombotic microangiopathy[14,15] and atherosclerotic vascular disease[2].

To date, the clinical detection of eGlx damage has relied on circulating shed eGlx fragments or side stream dark field imaging of sublingual micro vessels in conjunction with analysis software e.g. GlycoCheck™[16,17]. While these techniques have helped us confirm that eGlx damage occurs in human disease and is frequently systemic, they lack practicality for widespread surveillance in clinical practice[18,19]. We have recently published evidence showing that confocal microscopy can be used to measure eGlx damage in tissue sections[3,20-22]. During the development and validation of these light microscopy-based techniques we noticed that the red blood cells (RBC) within fixed tissue also have a complex surface glycocalyx (RBCGlx). We also observed that the RBCGlx depth in individual tissue biopsies appeared to mirror changes seen on the eGlx. Following these initial observations we have developed techniques to study the RBCGlx on peripherally sampled blood. Here we show the application of these techniques, using them

[1]Translational Health Sciences University of Bristol, Bristol, UK. [2]The Wolfson Bioimaging Facility, University of Bristol, Bristol, UK. [3]Sanford Burnham Prebys Medical Discovery Institute, La Jolla, USA. [4]North Bristol NHS trust, Bristol, UK. [5]University Hospitals Bristol, Bristol, UK. ✉e-mail: Matthew.Butler@bristol.ac.uk

to confirm that eGlx changes are 'mirrored' by changes on the RBC surface in health and disease. Investigating the underlying mechanism, we find this relationship is likely to be dependent on physical interaction between the cell types, confirming in vitro that interaction facilitates continual dynamic exchange of glycocalyx components.

## Results

### RBC have a glycocalyx that can be quantified using microscopy

Transmission electron microscopy (TEM) images of rat perfusion-fixed glomeruli labelled with Alcian blue (which binds to glycocalyx residues) suggested that RBC have a glycocalyx comparable in depth to adjacent eGlx (Fig. 1a). Biotinylated Lycopersicon esculentum lectin (LEL) was subsequently perfused into a donated healthy human placenta to adhere to glycocalyx before perfusion fixing the specimen for TEM. The lectin-biotin tag was then conjugated to quantum dots (Qdots™) before imaging their distribution using TEM. Figure 1b illustrates a maternal RBC within the placental circulation. Qdots™ confirmed that LEL, a lectin known to bind to the eGlx, bound to the human RBC surface and distributed uniformly within the RBCGlx structure.

To investigate the changes occurring on the RBC surface further we developed a blood smear-based protocol to use on peripherally sampled venous blood. This technique allowed us to measure the relative positions of the peak fluorescence signals from the RBCGlx (lectin, green) and the cell membrane (Octadecyl Rhodamine B Chloride (R18), red) to derive a measure of the RBCGlx depth.

Figure 1c is a confocal image of a single RBC optically sectioned through its widest axis with an illustrative line profile placed at 90 degrees to the membrane. A 3D reconstruction of a high-resolution z-stack labelled identically is shown in Supplementary Fig. 1A. The relative intensities of the light detected along the line profile are displayed (Fig. 1d). The distance between the peak lectin and R18 signals is referred to as the 'peak to peak' measure of Glx depth. Depth measurements are widely used to study eGlx changes in disease[3]. We have extensively validated this measure on eGlx using tissue samples and shown it to be superior to TEM imaging and measurement in predicting eGlx function[3].

To facilitate high throughput analysis of blood samples, we developed an automated, artificial intelligence assisted, software package. This software identifies all RBC in an image (Fig. 1e), places line profiles (every 15 degrees) around their circumference (Fig. 1f), measures fluorescence intensity along the line profile, before modelling Gaussian curves from the signals to minimise single pixel 'noise'. A wider field of view example of this identification image is shown in Supplementary Fig. 1B illustrating how the software ignores incompletely imaged RBC and platelets. A median 'peak to peak' measure for each RBC is then produced from the Gaussian curves. This method provides a sensitive measure of RBCGlx changes, capable of detecting very low-level enzymatic damage (Supplementary Fig. 1C, D). The 'peak to peak' measure is highly reproducible, minimising the impact of variations in laser power between microscopes, lectin batch or dilution (Supplementary Fig. 2).

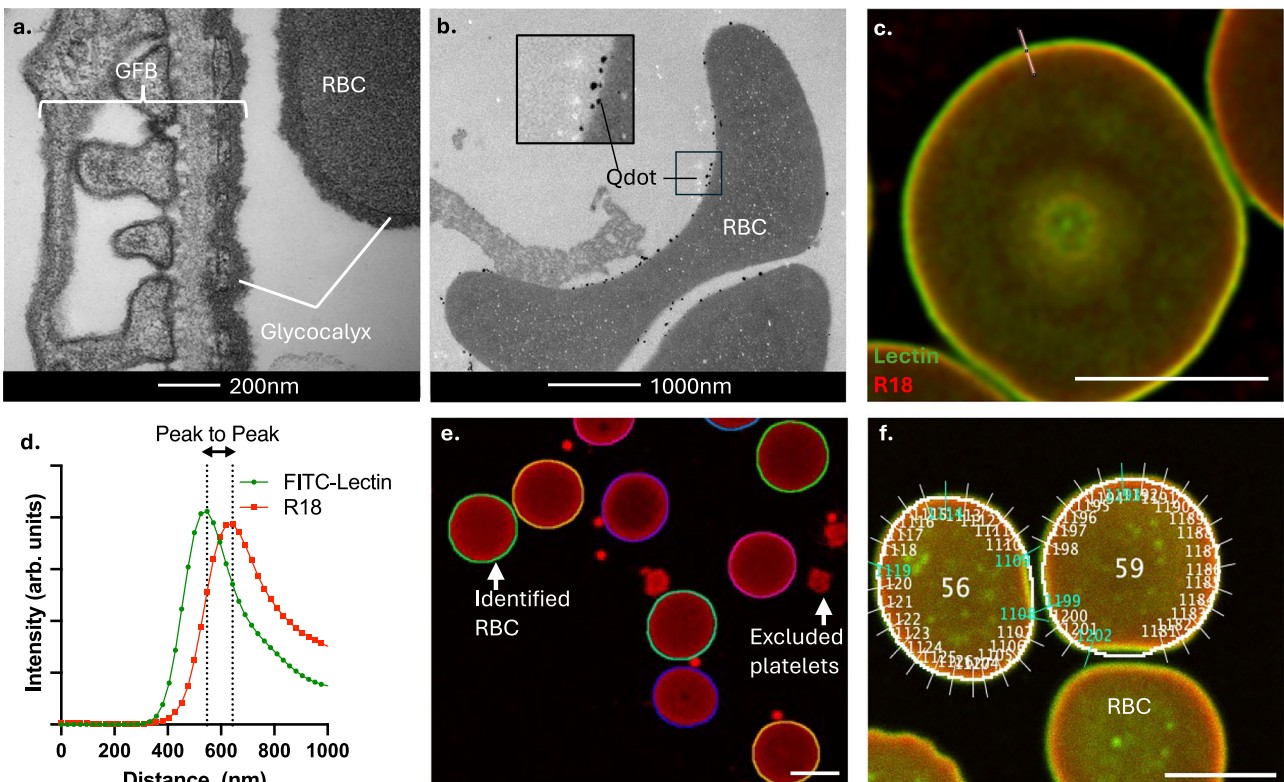

**Fig. 1 | RBC have a glycocalyx that can be quantified using microscopy. a** Alcian blue perfusion fixation and transmission electron micrograph (TEM) imaging of 5 healthy rats facilitated high resolution imaging of the rat glomerular filtration barrier and an entrapped RBC. Where RBC are present following perfusion, alcian blue bound to both the endothelial and RBC Glx allowing visualisation. **b** LEL lectin bound quantum dots perfused into a human placenta confirmed that lectins bind to the human eGlx and RBCGlx and distribute throughout the structures. **c** A confocal image of a healthy RBC labelled with FITC-LEL (green) and R18 (Red). The illustrative line profile at 90 degrees to the membrane was used to generate the illustrative 'peak to peak' measure of the RBC Glx thickness illustrated in figure (**d**) where distance (nano metres (nm) along the region of interest is plotted against measured intensity (arbitrary unit (arb. units)). of the two signals (green line = FITC-lectin, red line = R18). **e** Artificial Intelligence was trained to identify RBC on blood smears while ignoring adjacent platelets. **f** Identified RBC were examined automatically with line profiles measured every 15 degrees around the circumference of each RBC, white lines indicate included profiles, blue lines indicate automatically excluded measures. Scale bars = 5 μm unless indicated. Source data are provided as a Source data file.

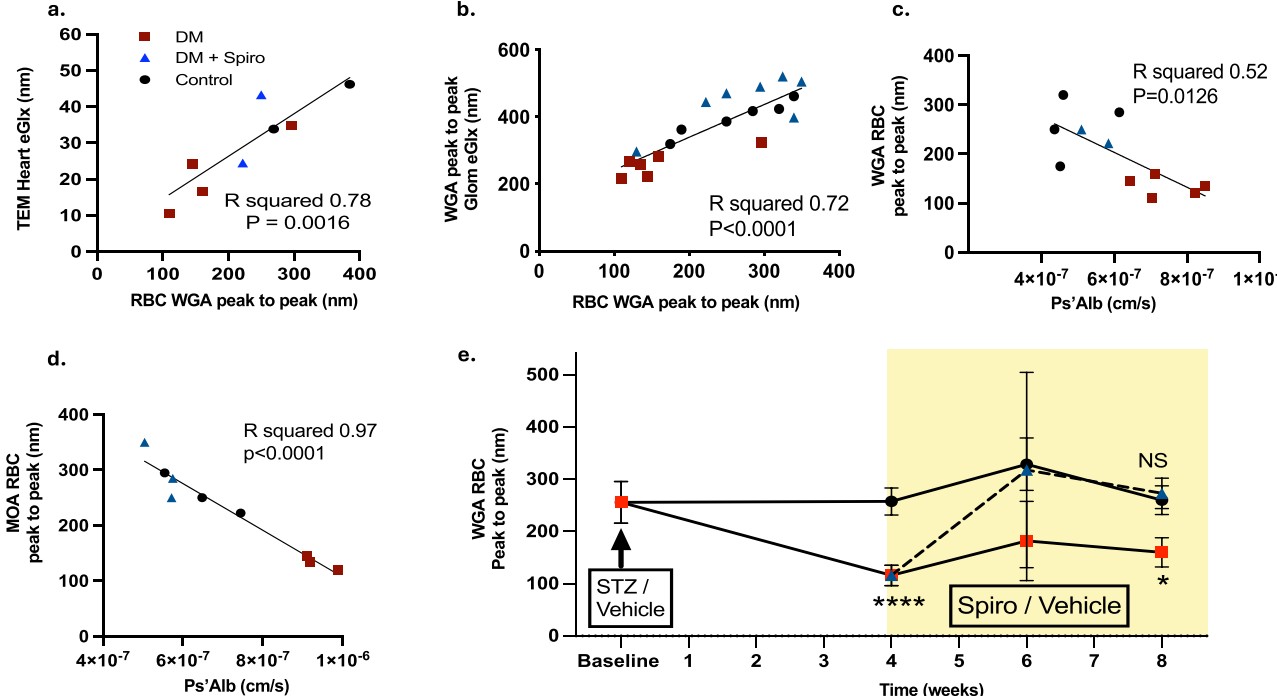

**Fig. 2 | RBCGlx damage predicts eGlx changes and endothelial function. a** We have previously studied eGlx changes and preservation using spironolactone using the illustrated protocol[1]. Alongside this published work we also measured RBCGlx changes. **a** RBCGlx peak to peak measures corelated with heart micro vessel eGlx depth (assessed using TEM/Alcian blue). Displayed data points = individual rats, analysis = linear regression. **b** In the same groups RBCGlx corelated with glomerular eGlx thickness measured using WGA lectin (week 6 onwards). **c** The RBCGlx depth measured using WGA lectin corelated with the ex vivo albumin permeability of glomeruli from the same animal. **d** The same effect was seen using MOA lectin to measure RBCGlx depth in a separate series of rats. **e** RBC sampling can be performed repeatedly. RBCGlx was shown to be impaired 4 weeks post STZ (diabetes) (two-tailed T-test comparison $n = 10$ control, 8 diabetic, $p < 0.0001$), however, glycocalyx repair following the initiation of Spironolactone could be detected after 2 weeks treatment and remained significant following 4 weeks of treatment (one-way ANOVA, $n = 6$ rats per group week 8, DM Spiro vs DM, $p = 0.033$), suggesting the RBCGlx could be used to monitor treatment responses to drugs targeting eGlx repair. Displayed data = mean +/− SEM per group, red squares = diabetic rats, blue triangles = diabetic rats treated with spironolactone, black circles = control rats. Source data are provided as a Source data file.

## RBCGlx damage predicts eGlx changes and endothelial function

Having established a reliable test for RBCGlx integrity we investigated if peripherally sampled RBC could be used to predict eGlx depth changes and variations in endothelial function. We have previously described eGlx damage in rat diabetic models and confirmed eGlx preservation using the mineralocorticoid receptor antagonist spironolactone[3]. In parallel we also measured RBCGlx changes (study protocol outlined in Supplementary Fig. 3). RBCGlx 'peak to peak' measured at week 8 post diabetes induction correlated significantly with the depth of the eGlx measured using TEM on capillaries from the left ventricle of the heart ($R^2 = 0.78$, $p = 0.001$ – linear regression) (Fig. 2a). RBCGlx 'peak to peak' measured at week 8 also significantly correlated with the thickness of the eGlx on glomerular capillaries ($R^2 = 0.72$, $p < 0.0001$ – linear regression) (Fig. 2b). The RBCGlx depth measured using Wheat germ agglutin (WGA) (Fig. 2c) or Marasmium oreades agglutinin (MOA) (Fig. 2d) lectins, predicted the albumin permeability of isolated glomeruli measured using a highly sensitive ex vivo glomerular permeability assay (linear regression $R^2 = 0.52$, $p = 0.12$ and $R^2 = 0.97$, $p < 0.0001$, respectively)[3,23]. This assay measures the functional integrity of the glomerular filtration barrier, including the eGlx on the surface of the fenestrated glomerular endothelium[23]. The use of RBCGlx to monitor systemic eGlx damage facilitated serial sampling (Fig. 2e). Serial blood sample analysis confirmed significant glycocalyx injury developed in diabetic rats by week 4 ($p < 0.0001$, ANOVA). In addition, we were able to detect improvements to the RBCGlx after 2 weeks of spironolactone treatment, which persisted until the experimental endpoint. These data show that the effect of

therapeutics targeting the endothelium and the eGlx can be monitored using peripherally sampled blood.

## Human RBCGlx depth predicts eGlx damage

We now sought to test if human RBCGlx could be used to assess eGlx changes. Anonomised surplus human kidney biopsy tissue (taken for a clinical indication but approved for subsequent research use with consent (approval reference H0102/45)) was labelled with Ulex europaeus agglutinin (UEA) lectin, R18 and 4′,6-diamidino-2-phenylindole (DAPI) as previously described[3]. Blinded eGlx depth measurements were taken on glomerular and peritubular capillaries contained within the renal cortex. Simultaneously blinded measures of the RBCGlx depth were taken manually from entrapped RBC within capillaries from each specimen. Figure 3a, b highlights that patients presenting with minimal change nephrotic syndrome (MCNS) (requiring a kidney biopsy) have significantly thinner eGlx and RBCGlx than patients with thin basement membrane disease (TMD). While it was not possible to analyse tissue from entirely healthy kidney (because kidney biopsies are not taken in the absence of a clinical indication) samples from patients with TMD were chosen because of their relatively preserved kidney function and identical collection and processing[3]. Comparing the median depth of RBCGlx on entrapped RBC with the eGlx depth measured on cortical capillaries demonstrated a close linear corelation (Fig. 3c).

To confirm if RBC sampled from peripheral veins could be used to predict eGlx measurements in patients we compared our RBCGlx depth measurements to data gathered using GlycoCheck™. GlycoCheck™ generates a measure of the perfused boundary region (PBR)

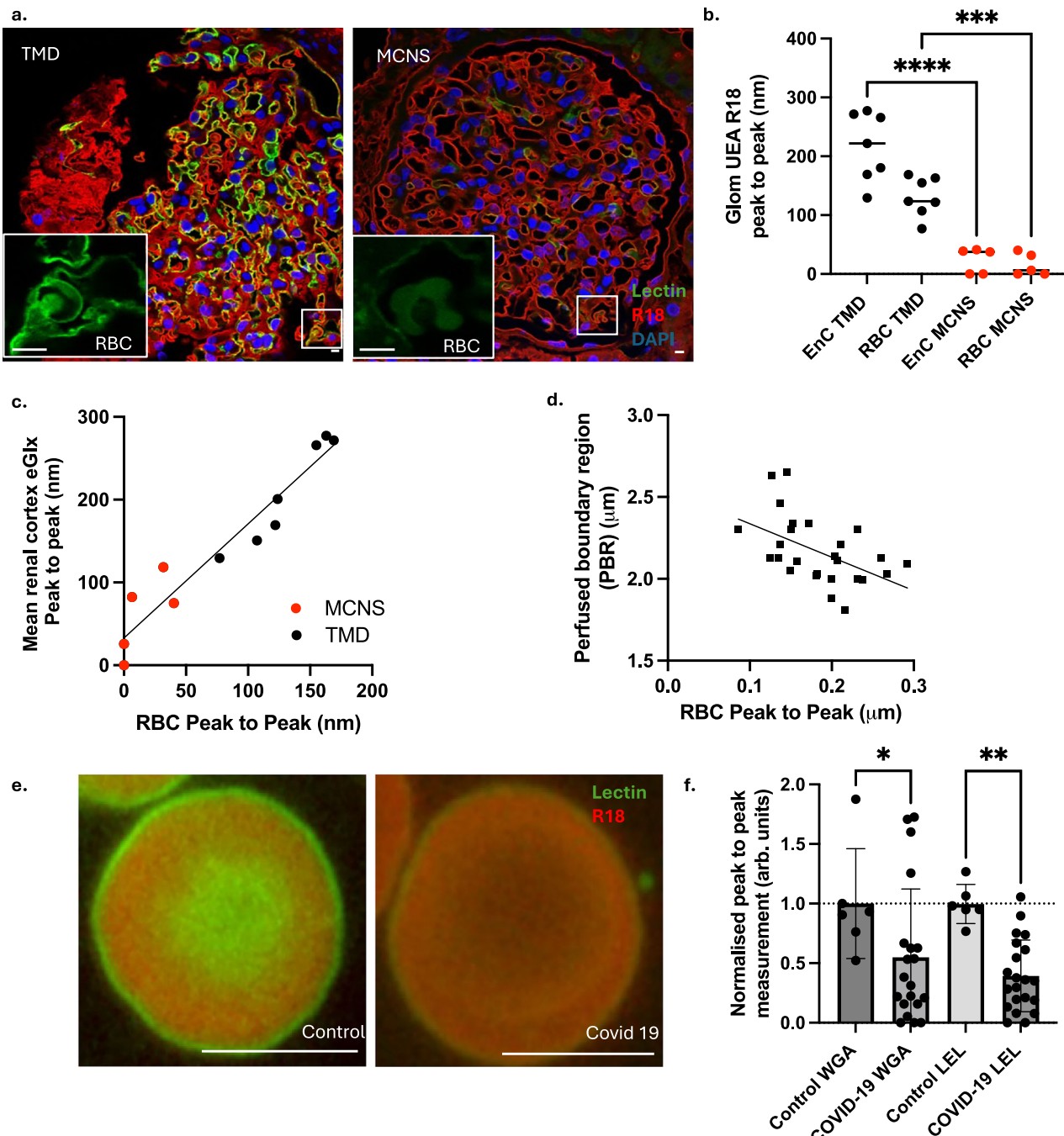

**Fig. 3 | Human RBCGlx depth predicts eGlx damage. a** Human kidney biopsies from patients with thin basement membrane disease (TMD) ($n = 7$) and minimal change nephrotic syndrome (MCNS) ($n = 5$) were labelled with R18 (red membrane label) and UEA lectin (green glycocalyx label). Patients with TMD had visibly more eGlx and RBCGlx compared to patients with MCNS. Inset images highlight the green channel on the indicated sections at higher magnification to highlight RBCGlx (scale bars 5 μm). **b** Peak to peak analysis of glomerular eGlx and entrapped RBC confirmed Glx loss in MCNS ($n$ = individual patients, ANOVA with Tukey correction, eGlx $p$ = <0.001, RBCGlx $p$ = 0.0003). (black circles = TMD, red circles = MCNS). **c** Linear regression confirmed a close linear correlation between the depth of entrapped RBCGlx and capillary eGlx within the renal cortex ($r^2 = 0.9226$, $p < 0.0001$) (black circles = TMD, red circles = MCNS). **d** In healthy pregnant women

($n = 27$) a significant correlation (linear regression) exists between the PBR measurement in micro vessels (generated using GlycoCheck™) and the RBCGlx depth across all vessel sizes 5–25 μm ($R^2 = 0.25$, $p = 0.007$) and for each size range sampled by GlycoCheck™ (5–9 μm $r^2 = 0.21$, $p = 0.02$, 10–19 μm $r^2 = 0.16$, $p = 0.04$, 20–25 μm $r^2 = 0.30$, $p = 0.007$), confirming that RBC and endothelial Glx structures 'mirror' each other in humans. **e** Representative wide field images of a heathy RBC labelled with R18 (red) and LEL lectin (green) and an RBC from a patient hospitalized with COVID-19 labelled and imaged blind using identical methods (Scale bars 5 μm). **f** RBCGlx depth measured using WGA or LEL both confirm significant changes in the RBCGlx between controls ($n = 6$) and patients with COVID-19 WGA ($n = 20$, $p = 0.0452$), LEL ($n = 21$, $p = 0.0032$, Kruskal–Wallis analysis. Graph illustrates normalized mean, +/− SD). Source data are provided as a Source data file.

in sublingual micro vessels. This measure is low when individuals have a thick eGlx and becomes higher if the eGlx is damaged. The PBR measurement is one of the most widely used tools for assessing human eGlx integrity currently[18,24]. Blood smears were prepared from venous blood samples taken from pregnant subjects at the booking visit (first trimester) as part of a 'Study in Pre-eclampsia And Diabetes aetiology (SPADE)' a prospective observational study of women attending antenatal care in Bristol (UK) (approval reference 06/Q2006/54). Simultaneously GlycoCheck™ was performed to assess the eGlx thickness. Figure 3d illustrates the significant linear corelation between the RBCGlx thickness and the sublingual PBR measured in micro vessels (5–25 µm ($R^2 = 0.25$, $p = 0.007$ – linear regression) confirming that the RBCGlx mirrors variations in eGlx depth variations in this population. We found the RBCGlx depth corelated with the PBR in all analysed vessel size categories in this cohort (Supplementary Fig. 4) suggesting RBC can 'report' changes occurring in all blood vessels sizes between 5 and 25 µm in diameter. Analysis of a subset of clinical samples from the SPADE study (labelled identically with a single batch of lectin and imaged using fixed settings on a single microscope blind) provided measurements of lectin florescence intensity within the RBCGlx. RBC surface lectin intensity corelated with PBR measurements ($R^2 = 0.23$, $p = 0.049$) but the correlation was weaker than that seen with 'peak to peak' measurement in the same cohort ($R^2 = 0.52$, $p = 0.001$) (Supplementary Fig. 5).

To facilitate subsequent clinical studies (where delays in processing blood samples were unavoidable), we utilised Transfix® filled EDTA vacutainers for sample collection and temporary storage. Transfix® stabilises the RBCGlx at the point of collection, facilitating stable storage at 4 degrees centigrade[25]. To test if we could detect pathological changes to the RBCGlx in the context of clinical research, RBC samples were collected as part of the DIagnostic and Severity markers of COVID-19 to Enable Rapid triage (DISCOVER) study with a working hypothesis that COVID-19 infection would be associated with significant eGlx damage[26,27]. This study prospectively recruited participants from a single UK hospital from 30.03.2020 (approval reference 20/YH/0121). Summary demographic details of the enrolled subgroup of participants where the RBCGlx was measured are presented in Supplementary Fig. 6. Comparing blood samples from hospitalised patients with COVID-19 and control participants suggested marked RBCGlx damage in COVID-19 patients (Fig. 3e). Subsequent analysis of blood samples labelled with either WGA or LEL lectins, confirmed a significant reduction in RBCGlx thickness (Kruskal–Wallis $p < 0.05$ and <0.005, respectively) (Fig. 3f) consistent with subsequent publications[24]. Importantly these data also confirm that RBCGlx assessment can be successfully integrated into clinical research protocols.

Having identified the utility of the RBCGlx we sought to investigate the underlying mechanisms behind the close relationship seen between the RBCGlx and eGlx in the systemic vasculature using in vitro models.

### Direct contact with endothelial cells can restore the RBCGlx

Neuraminidase removes sialic acid (SA) residues from glycoproteins. Neuraminidase (300 mU/ml) resulted in a progressive reduction in WGA binding consistent with loss of SA from the RBCGlx (Fig. 4a). Marked RBCGlx loss was seen after 60 min exposure (Fig. 4b). However, following 300 mU/ml neuraminidase exposure for 1 h, RBC WGA binding could be restored by 'flowing' RBC over healthy endothelial cell monolayers (5 dyn for 6 h) (Fig. 4c) (ANOVA $p < 0.0001$).

Neuraminidase, by removing terminal SA residues, increases the availability of binding sites for wisteria lectin (WL) and peanut lectin (PNL) which are otherwise blocked by SA[28]. Treatment of RBC with low dose neuraminidase (30 mU/ml) significantly increased FITC-WL and FITC-PNL binding assessed by peak fluorescence intensity at the RBC membrane (Fig. 4d, e). However, interacting washed RBC previously

exposed to 30 mU neuraminidase, with healthy eGlx significantly reduced subsequent WL binding ($p < 0.001$ ANOVA) and PNL binding ($p < 0.001$ ANOVA). These data collectively suggest that direct RBC–endothelial interaction can alter the RBCGlx composition by altering terminal SA expression. In contrast, maintaining RBC in endothelial-conditioned media consistently had no effect on the RBCGlx suggesting that RBC have limited capacity for auto recovery, and that soluble mediators in the culture media were not responsible for RBCGlx recovery.

### Changes in endothelial heparan sulphate expression can be detected on RBC

Heparan sulphate (HS) is a key component of the eGlx. The expression of HS within the eGlx has been shown to alter during the development of diabetes[20]. We therefore sought to investigate if altering HS expression within the eGlx could alter the RBCGlx.

Treating RBC with the enzyme heparinase III (1 Sigma U/ml) significantly reduced RBCGlx LEL lectin binding consistent with the expression of the enzyme's specific substrate HS on the RBC surface (Supplementary Fig. 7) and consistent with previous publications[29]. Exotosin-1 (EXT1) is a rate-limiting enzyme essential for the synthesis of HS. Overexpression of EXT1 in endothelial cells increased HS within the eGlx (Supplementary Fig. 8). We therefore investigated if forced production of this key eGlx component by means of EXT1 overexpression in endothelial cells could enhance the 'recovery' of RBCGlx.

The glycocalyx was depleted on isolated RBC using 300 mU/ml neuraminidase (Fig. 5a). RBC were then interacted with wild type or EXT1-overexpressing endothelial cells (5 dyn, 1 and 14 h). Directly interacting glycocalyx-depleted RBC with wild type endothelial cells for 1 h had no measurable effect on RBCGlx recovery (Fig. 5a). However, after 14 h, cell-cell interaction resulted in significant RBCGlx recovery (increase in 'peak to peak' measure, Kruskall-Wallis, $p < 0.005$) (Fig. 5b). EXT1-overexpressing endothelial cells (Ext1 +) expedited the recovery of RBCGlx (significant recovery after 1 h (Kruskall-Wallis, $p < 0.0001$)) (Fig. 5a). After interaction with either control or EXT1+ cells for 14 h, no significant damage to the RBCGlx remained detectable and no significant differences remained between the groups suggesting saturation of the recovery process. Again, RBC incubated in endothelial cell 'conditioned media' for 14 h demonstrated no glycocalyx recovery (Fig. 5b). Representative images taken following 1 h interaction illustrate variations seen in the RBCGlx between the conditions (Fig. 5c, d). Representative images for all conditions are shown in Supplementary Fig. 9.

Having established that the eGlx can donate components to RBCGlx in vitro, we sought to investigate if this effect was significant in vivo. To establish this, we utilised an endothelial cell-specific EXT1 inducible knockout mouse model generated by crossing Cdh5CreERT2 (Taconic Biosciences) mice with EXT1fl/fl mice[20,30]. Endothelial EXT1 knockout mice (EXT1– (KO)) are healthy but display reduced HS within their eGlx and have significant alterations in endothelial barrier function[20]. We hypothesised endothelial specific knockout of EXT1 would result in changes to the RBCGlx. RBC LEL lectin intensity was visibly reduced in EXT1- mice relative to littermate controls (Fig. 6a, b). RBC LEL surface intensity measurements and 'peak to peak' analysis confirmed RBCGlx loss Fig. 6c (t-test, $p < 0.005$) and Fig. 6d (Kruskal–Wallis, $p < 0.0.1$). These data suggest that the transfer of sugars from endothelial cells to RBC could be important in maintaining the RBCGlx in circulation.

### RBC can donate glycocalyx components to endothelial cells

Having confirmed that endothelial cells can donate glycocalyx components to RBC, we sought a method to investigate whether a reverse relationship existed. Endothelial cells have the capacity to rapidly regenerate glycocalyx components once removed limiting the utility of simple transfer experiments. We therefore looked to label eGlx

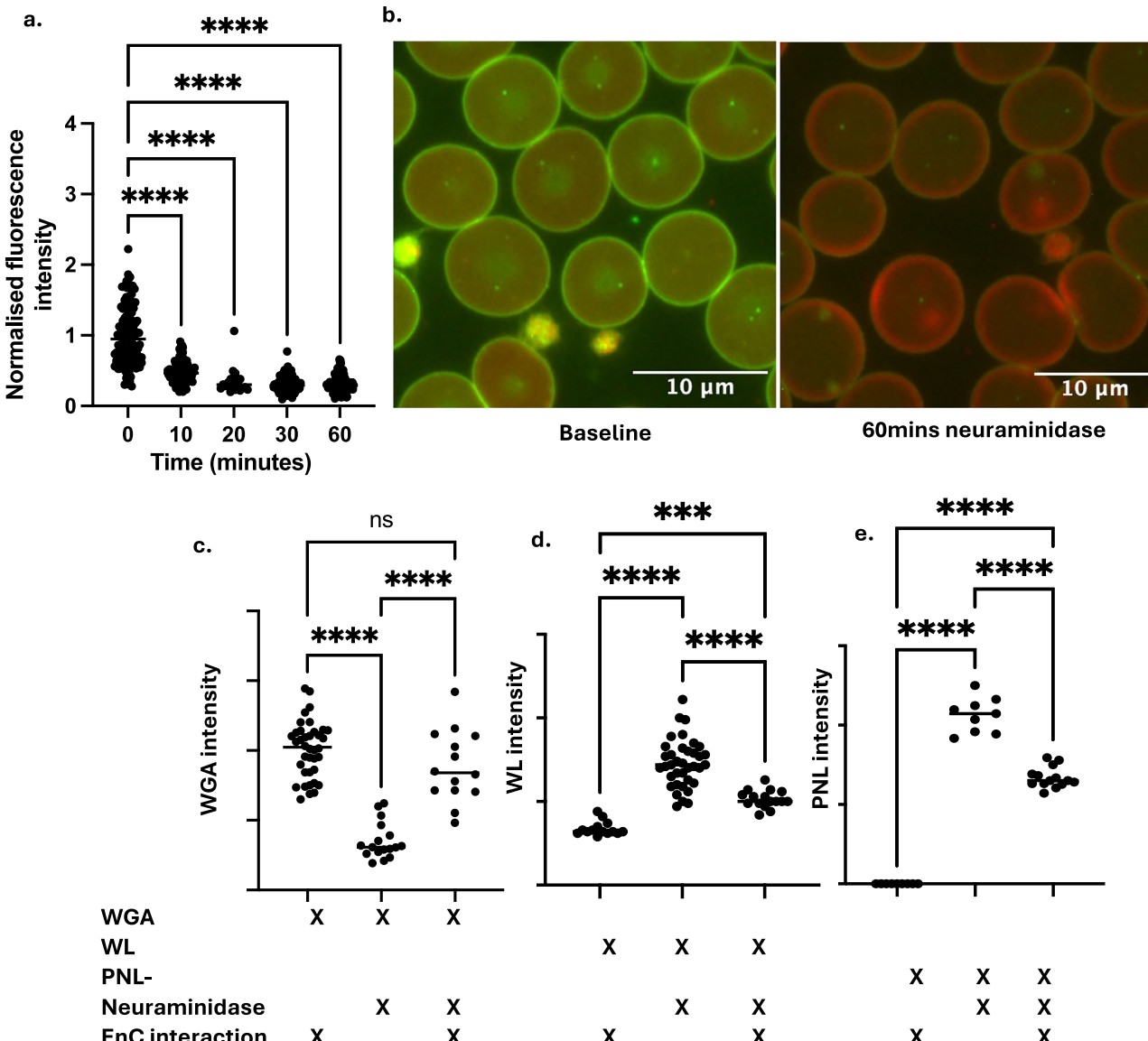

**Fig. 4 | Direct contact with endothelial cells can restore the RBCGlx.**
**a** Neuraminidase (300 mU/ml) reduced RBC FITC-WGA binding measured on blinded images relative to baseline values (one-way ANOVA $p < 0.0001$, Dunnett's multiple comparisons test (to control) confirmed all time points resulted in significant damage ($p < 0.0001$) (displayed $n$ = RBC median values, 3 experimental repeats). **b** Representative images confirmed RBCGlx loss and that RBC remained intact after 1 h exposure to neuraminidase. **c** Endothelial interaction ($n = 14$) restored RBC WGA binding to normal levels after exposure to 300 mU/ml neuraminidase (ANOVA, with Tukey's multiple comparison test to control ns $p = 0.138$), (Tukey's comparison to post neuraminidase $p < 0.0001$), suggesting RBCGlx repair. RBC flow in the presence of endothelial conditioned media (but no endothelial cells) had no effect (Supplementary Fig. 4). **d** RBC exposed to neuraminidase (30 mU/ml 1 h) display significantly increased wisteria lectin binding (WL) (ANOVA, with Tukey's multiple comparison tests, $p < 0.0001$). However, endothelial interaction ($n = 14$) for 6 h significantly reduced FITC-WL ($p = 0.0001$). **e** RBC exposed to neuraminidase (30 mU/ml 1 h) display significantly increased peanut lectin (PNL) binding compared to control RBC ($p < 0.0001$). Endothelial interaction ($n = 14$) for 6 h significantly reduced PNL surface intensity ($p < 0.0001$). WFA and PNL data collectively suggest removal or 're-shielding' of the relevant lectin binding sites through interaction with eGlx (**c–e** displayed $n$ = RBC mean per replicate, $p < 0.05$ *, $p < 0.01$ **, $p < 0.005$ ***, $p < 0.0001$ ****). Source data are provided as a Source data file.

components before interacting endothelial cells with RBC and studying component transfer. Initial experiments utilised LEL and WGA lectin. The lectins were bound to the RBCGlx before extensive washing and subsequent endothelial monolayer interaction. Lectins with fluorescent tags rapidly transferred onto the endothelial surface and the transfer appeared to be increased when monolayers were enzymatically damaged or exposed to diabetic conditions before interaction (Supplementary Figs. 10 and 11). However, proving the lectin (and the FITC tag) remained attached to eGlx components and were not influencing the transfer process was not possible. We therefore utilised 'Click' chemistry to confirm the transfer of glycocalyx components and

interrogate this relationship further under physiological conditions. Adding N-azidoacetylmannosamine (ManNAz) to the endothelial culture medium resulted in the integration of an azide-modified mannosamine into sialic acids (SA). This labelled sugar could then be exploited by a ligation or 'Click' reaction between the azide and an alkyne to conjugate a fluorophore to all substrates in which the Man-NAz sugar is incorporated (Fig. 7a)[31]. High resolution z-stack series confirmed ManNAz integrated into the eGlx on the endothelial cell surface (Fig. 7b). Enzymatic removal of eGlx further confirmed that ManNAz integrated into the eGlx on the cell surface (Supplementary Fig. 12). RBC that interacted with endothelial cells exposed to control

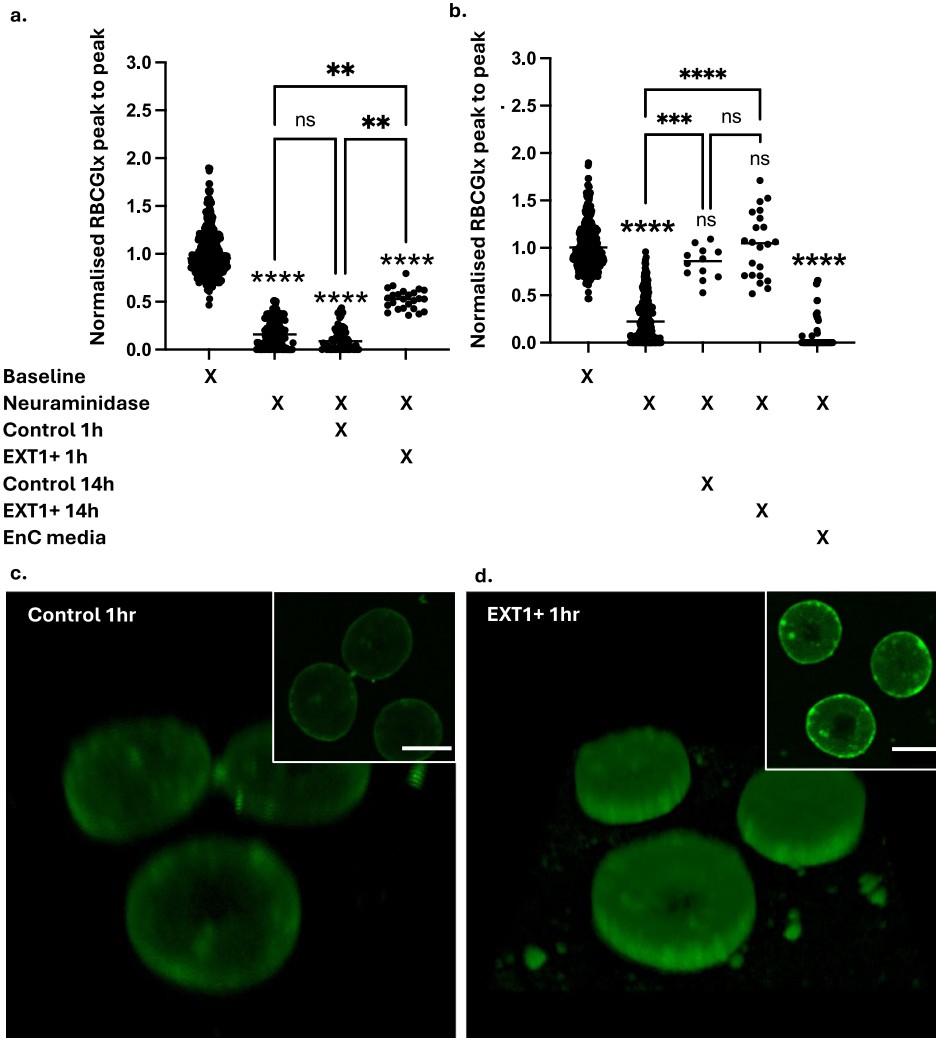

**Fig. 5 | Changes in endothelial heparan sulphate expression can be detected on RBC. a** Following neuraminidase (30 mU/ml 1 h) LEL lectin peak to peak thickness was significantly reduced on human RBC (ANOVA, $p < 0.0001$, Kruskal–Wallis test control vs enzyme baseline $p < 0.0001$). 1 h interaction between enzyme depleted RBC and control endothelial cells had no measurable effect ($p > 0.9999$). In contrast, 1 h interaction between enzyme depleted RBC and EXT1 over-expressing endothelial cells (EXT1 +) significantly restored the RBCGlx ($p = 0.0087$). **b** After 14 h interaction, significant recovery of the RBCGlx was seen (neuraminidase vs control 14 h $p = 0.0079$, (neuraminidase vs EXT1+ $p = 0.0004$), but no significant residual effect from EXT1 over expression was detected between the groups ($p > 0.999$). Circulating RBC in endothelial cell-conditioned media for 14 h had no effect on the RBCGlx thickness. **c** Representative 3D image of RBC labelled with LEL lectin (green) following neuraminidase and 1 h interaction with control endothelial cells. The inset displays single z stack image of the same cells and a 5 μm scale bar. **d** Representative 3D image of RBC labelled with LEL lectin (green) following neuraminidase and 1 h interaction with endothelial cells over expressing EXT1. The inset displays single z stack image of the same cells and a 5 μm scale bar. Both images (**c**) and (**d**) were taken while blinded using fixed settings. The full series of representative images are shown in Supplement Fig. 8 ($n$ = RBC median values from 3–5 experimental repeats, Kruskal–Wallis analysis used for comparisons with no distribution assumption ($p < 0.05$ *, $p < 0.01$ **, $p < 0.005$ ***, $p < 0.0001$ ****). Source data are provided as a Source data file.

media containing DMSO (vehicle) before the 'Click' reaction expressed minimal non-specific surface staining (Fig. 7c). In contrast, RBC interacted with washed endothelial cells previously cultured in media containing ManNAz bound a significant quantity of the alkyne-conjugated fluorophore (green) indicating significant transfer of azide modified sialic acid onto the RBC surface (Fig. 7d). Intensity measurements at the RBC surface confirmed this effect (Fig. 7e) (t-test, $p < 0.001$).

Separately, control RBC (interacted with control endothelial cells) and RBC (loaded with 'Click'-labelled eGlx through endothelial interaction for 24 h) were washed and transferred onto fresh endothelial cell monolayers (never exposed to ManNAz). After 4 h interaction (5 dyn) RBC were removed, and the endothelial cells were washed and labelled using standard protocols. Minimal (non-specific) surface 'Click' label was seen on endothelial cells exposed to control RBC

(Fig. 7f). However, after 4 h interaction with 'Click'-loaded RBC, patches of 'Click'-labelled glycocalyx were visible on the new endothelial monolayers (Fig. 7g). With the microscope set to ensure no background signal on control endothelial monolayers the median intensity from endothelial cultures (by coverslip) after exposure to labelled RBC was 8.9 (arbitrary unit measure of mean fluorescence intensity). A one sample t-test and Wilcoxon test confirmed significant transfer of the ManNAz labelled eGlx onto the 'clean' endothelial cells via the RBC ($p = 0.002$) (Fig. 7h). Again, transfer of media exposed to washed 'Click'-labelled cells had no effect on the fresh endothelial monolayer labelling.

Together these data suggest that bidirectional transfer of glycocalyx sugars occurs between endothelial cells and RBC when the two cell types are in direct contact. This relationship is likely to contribute to the close 'mirroring' effect seen in vivo between the eGlx and

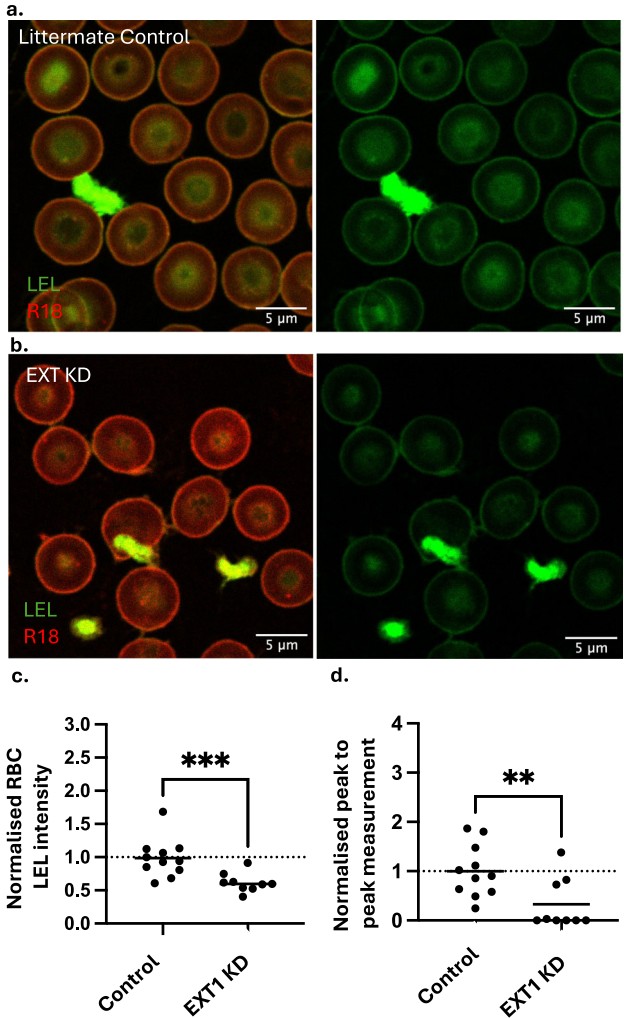

**Fig. 6 | RBC from mice with endothelial specific deletion of heparan sulphate have reduced RBCGlx. a**, **b** Blood smears prepared from litter mate control mice (**a**), and mice with conditional endothelial specific EXT1 knock down (EXT KD) (**b**). Both slides were labelled identically with R18 (red cell membrane) and LEL lectin (green glycocalyx) and were imaged blind, the combined image and identical isolated green channel are displayed. The Images highlight visible differences in RBCGlx, with visible RBCGlx loss seen in EXT1 KD mice, an effect not seen on platelets or leucocytes. **c** RBC surface LEL lectin intensity was significantly reduced in EXT1 KD mice ($n = 9$) compared to littermate controls ($n = 10$) (two-tailed Mann–Whitney, $p = 0.0008$). **d** RBC Glx was significantly thinner in EXT1 KD mice ($n = 9$) compared to littermate controls ($n = 10$) (two-tailed Mann–Whitney, $p = 0.0093$). Displayed $n =$ individual mice ($p < 0.01$ **, $p < 0.005$ ***). Source data are provided as a Source data file.

RBCGlx thickness in both rodent disease models and human observational studies.

## Discussion

We have demonstrated that the RBCGlx mirrors changes occurring simultaneously on the eGlx within the kidney, heart and peripheral microvasculature. Investigating the underlying mechanism responsible for this effect we identified the bidirectional transfer of glycocalyx components between the RBCGlx and eGlx. Subsequent investigations have confirmed specific transferred components. Applying these discoveries we have shown that RBCGlx depth measurements can predict endothelial function in rodent models. In human disease our data from patients with COVID-19 agree with the

published literature (suggesting eGlx damage occurs early on in this condition), but they represent the first direct measurements of glycocalyx damage in COVID-19 to our knowledge[24,25].

The importance of the eGlx in maintaining endothelial function is well established[2,4,14,20,32]. However, the eGlx is damaged in human disease and this damage frequently occurs early in the disease course[2,3]. In diabetic animal models eGlx damage is detectable before albuminuria develops suggesting that eGlx damage pre-dates irreversible microvascular injury[3]. We have shown that targeted interventions to preserve the eGlx at this early stage, delay the onset of microvascular damage[3,20]. As we move towards individualised medicine, detecting early microvascular injury in the form of eGlx loss could therefore identify the subset of patients that will benefit most from targeted interventions.

In developing methods to measure the RBCGlx we have sought to utilise lectins that bind uniformly within the Glx and are unaffected by blood group[33,34]. Species variations also contributed to these decisions, we found MOA lectin to bind strongly to rodent RBC and eGlx, however (as predicted), binding was limited in humans due to the lack of Gal alpha(1,3)Gal-containing sugar expression[35]. The application of AI and automated computer analysis of images has allowed us to measure RBCGlx with great reproducibility despite the limited absolute resolving power associated with light microscopy. Key to these techniques is the use of the cell membrane as a reference point and the application of Gaussian models to reduce noise in the generated light signals[3]. These elements, combined with the large number of measurements made on each sample, ensure that even subtle changes induced by very low-level enzyme exposure can be seen. It seems likely therefore that even damage in early disease will be detectable using these techniques. The comparison of RBCGlx assessment methods (RBC surface lectin intensity and 'peak to peak' based measures) currently suggests that 'peak to peak' assessment mirrors variations detected using GlycoCheck™ more closely. Because RBCGlx 'peak to peak' and GlycoCheck™ measures both report glycocalyx depth changes this is not unexpected. Moving forward intensity-based measures will need to be validated against the more established use of glycocalyx depth changes, but combining intensity measures with the depth measurements could prove useful when looking for early subtle glycocalyx damage. In addition, the relative simplicity of quantifying glycocalyx damage using single binding molecules (and in this case an intensity-based measurement) make this an attractive area for future research. If perfected RBCGlx intensity-based measures could facilitate rapid testing for glycocalyx damage in patients potentially making them a valuable tool for clinicians managing acute illnesses such as sepsis. RBC changes have previously been shown to predict disease states including hypertension[36], preeclampsia[37], kidney disease[38], and diabetes where the degree of RBC charge loss predicts retinopathy grade and glomerular albumin permeability[39,40]. However, the mechanisms behind these associations have not previously been elucidated. Human RBC typically circulate for between 70 and 140 days[41]. The negative charges on RBC and endothelial cells are insufficient to limit direct contact between their glycocalyces, as a result when damaged endothelial monolayers interact with RBC they have been reported to damage the RBC surface[42,43]. We propose that some of the reciprocal changes seen in historical reports (where RBCGlx damage was induced following interaction with enzymatically damaged eGlx) could have been the result of the RBC donating glycocalyx components to the damaged endothelium. Circulating levels of enzymes, capable of removing key elements of the RBCGlx, are also known to increase in multiple diseases[11,29,44,45]. RBC lack significant synthetic capacity[46]. However, evidence exists that de-sialated young RBC can be 'repaired' in the liver and returned to the circulation[47]. During infection individual RBC have also been shown to dynamically alter their SA expression acting as both SA donors and recipients to each other[48]. RBC also adsorb blood group antigens onto their surface during circulation[49],

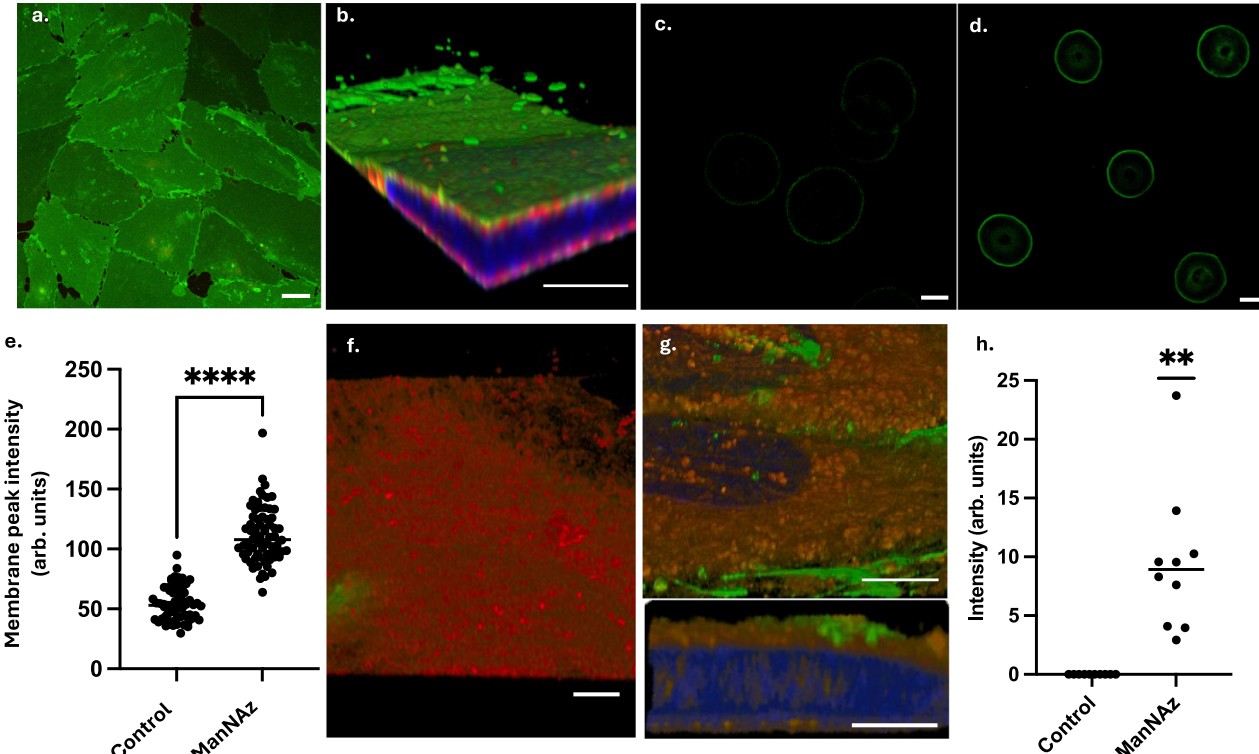

**Fig. 7 | RBC can donate glycocalyx to endothelial cells. a** Confocal imaging of endothelial monolayers highlights universal ManNAz integration. **b** High resolution confocal imaging confirms ManNAz labelled sialic acids (green) are expressed on the endothelial cell surface external to the cell membrane (red, R18) and cell nucleus (blue DAPI). **c** RBC exposed to control endothelial cells for 24 h (not fed ManNAz but processed with Click™ fluorophore (green)) show minimal background florescence. **d** RBC exposed to ManNAz labelled endothelial cells display the Click™ binding site on their surface following interaction suggesting transfer. **e** Quantification of individual RBC Click™ signal intensity at the membrane confirms eGlx – RBCGlx sialic acid transfer (biological $n = 10$, two-tailed t-test $p < 0.0001$).

**f** Endothelial cells exposed (4 h) to RBC previously interacted with control endothelial cells for 24 h demonstrate no significant 'click' binding (green). **g** Endothelial cells exposed (4 h) to RBC 'loaded' with 'click' labelled eGlx exhibit dense patches of click labelled glycocalyx – suggesting RBC can redistribute glycocalyx components from one endothelial population to another, inset highlights transferred material (green) remains on the cell surface. **h** Quantification of green 'click' label intensity on cultured endothelial monolayers after RBC contact confirmed significant transfer from 'loaded' RBC onto the endothelial surface ($n = 10$) (one sample t and Wilcoxon test ($p = 0.0020$)) (scale bars 5 μm ($p < 0.01$ **, $p < 0.005$ ***, $p < 0.0001$ ****)). Source data are provided as a Source data file.

and 'donate' functional GPI-anchor linked proteins to the vascular endothelium[50] and to each other[51,52]. In our experiments we included conditioned media controls to determine whether eGlx components shed into the culture medium could repair the RBCGlx. These experiments confirmed that in the absence of direct interaction no RBCGlx repair is seen. In contrast, by manipulating RBC SA and carefully interrogating the recovery following interaction, we have confirmed that SA residues transfer onto the RBC to 're-mask' exposed lectin binding sites only following direct interaction between the cell types. Using 'Click' chemistry, we demonstrate that the RBCGlx can donate components back to the eGlx. This intimate relationship results in continual exchange and may explain how RBC can maintain a glycocalyx in the circulation. The level of interaction required for transfer events to occur is currently unknown but further modelling could be employed to explore this process further. Our data suggest that glycocalyx exchange continues in diabetic disease models but again whether other pathologies could alter the transfer process is currently unknown. Now that we are beginning to understand the extent and potential importance of glycocalyx transfer new methods are needed to study it in human disease to accelerate this emerging field of research. Investigating the underlying physiology we have focussed on HS, a highly sulphated glycosaminoglycan and a major component of the eGlx[20]. HS is a known biding site for malarial parasites and has been shown to also be expressed on the RBC surface[29]. In vitro we confirmed that endothelial cells overexpressing HS can repair the RBCGlx more rapidly than control endothelial cells. In vivo endothelial-specific

inactivation of HS synthetic capacity (endothelial EXT1 knock down) resulted in significant RBCGlx loss. These data together suggest that endothelial cells play an active role in maintaining the RBCGlx via the expression of HS.

Simultaneously we found that the intimate relationship between the RBCGlx and eGlx provides a means of real-time assessment of the eGlx response to disease and therapeutic interventions. The RBCGlx depth on peripherally sampled venous blood corelated with eGlx depth assessed directly in animal models using TEM and light microscopy, and indirectly in humans using GlycoCheck™. We believe using RBC to monitor eGlx changes holds significant advantages over existing techniques. GlycoCheck™ is reliant on the localised motion of RBC in the circulation, thus subtle changes in BP and microvascular flow (after caffeine or food intake for example) may affect the readings. Trained operators are also needed and for the patients GlycoCheck™ is time consuming[18]. The use of shed glycocalyx component levels is also indirect and likely to be relatively unreliable in chronic disease because the individual plasma or urine levels provide a composite measure influenced dynamically by the rate of production, shedding and clearance of individual glycocalyx components.

Although we have shown that SA and HS are key contributors to the transfer of glycocalyx we do not yet know the full extent to which sugars are exchanged. Future work will address this question systematically. We are also aware that the eGlx composition varies from tissue to tissue and we do not yet know if the RBCGlx reflects changes at all sites. Data gathered to date, however, suggests that the RBC provides a

readily accessible means through which we can assess systemic vasculature eGlx integrity thus providing researchers with a 'liquid biopsy'. We hope this work will stimulate others to investigate the importance of the glycocalyx on endothelial cells and RBC and accelerate this important field of research towards integration in clinical practice.

## Methods

### Human blood

Studies were planned, conducted and reported in accordance with the declaration of Helsinki 2013. All human blood samples were collected with full written consent. Sample collection from women enrolled in the Study in Pre-eclampsia And Diabetes Aetiology (SPADE) a prospective observational antenatal study in Bristol (UK) was approved by the National Research Ethics Service (NRES) Southwest – ref 06/Q2006/54. Blood samples were collected into lithium heparin (LiH) blood vacutainers, stored at 4 degrees and processed within 4 h. Samples collected as part of the DIagnostic and Severity markers of COVID-19 to Enable Rapid triage (DISCOVER) study were taken prospectively from recruited participants in a single UK hospital (ethics approval via South Yorks REC: 20/YH/0121). For This study transfix™ EDTA vacutainers were utilised. Samples were stored at 4 degrees after 2 h at room temperature and processed within 5 days. After images were obtained all samples were disposed of in accordance with local guidance.

For in vitro glycocalyx transfer experiments 2 million RBC per replicate were isolated from heathy donor whole blood samples using acrodisc syringe filters (Pall, AP 4851). Samples were washed by suspending isolated RBC in sterile phosphate buffered saline, or EBM2 cell culture media for 5 min before centrifuging at $200 \times g$ (no brake and slow ramp) for 5 min to pellet the RBC. This process was repeated 3 times. Blood smears with DAPI staining confirmed removal of all visible leucocytes.

### Animal blood

Details of the diabetic rat model are provided in Supplementary Fig. 3. Male Wistar rats (150–200 g, Charles River Laboratories) were maintained in a conventional facility (21–24 °C and 12:12 h light/dark cycle). Randomized rats were injected intraperitoneally with 50 mg/kg STZ (S0130; Sigma-Aldrich) (25 mg/mL in 10 mM sodium citrate, pH 4.5). Four weeks after STZ injection, spironolactone (spiro) was given for 28 days (S3378; Sigma-Aldrich) at 50 mg/kg made up in corn oil (C8267; Sigma-Aldrich), and rats were culled at week 8 after STZ injection. Comparisons were made with vehicle-treated rats (10 mM sodium citrate [pH 4.5]). Glycemia, by tail-tip blood droplet analysis using a glucometer (Accu-Chek Aviva; Roche), was measured 3 weeks after STZ, and rats with glycemia ≥15 mmol/L were considered diabetic and included in the study. Blood was sampled from Wistar rats' tail vein (50 µl) using standard aseptic techniques. Blood smears were prepared and fixed in 100% methanol using standard techniques.

Endothelial specific *EXT1* conditional knockout mice (*EXT1(ECKO)*) and litter mate control mice were generated on a C57BL/6 background by crossing Cdh5(PAC)-Cre^ERT2 mice (Taconic) with *EXT1*-floxed (*EXT1fl/fl*) mice and maintained in a conventional facility (21–24 °C and 12:12 h light/dark cycle)[30]. In male Cdh5(PAC)-Cre^ERT2;EXT1^flox/flox mice (aged 6–10 weeks) gene excision was induced by tamoxifen IP injection (75 mg/kg) for 5 consecutive days. Blood samples from Cdh5(PAC)-Cre^ERT2;EXT1^flox/flox mice and their littermate controls were taken under terminal anaesthesia. All animals were kept according to the *Guidelines on the use of Animals in research* and all procedures performed under licence and approved by the UK Home Office.

### Human kidney

Human kidney biopsy samples were archived anonymized samples taken for a clinical indication. Written informed consent was given by each participant for anonymised surplus tissue (not needed for clinical diagnosis) to be used in research. Ethical approval was granted via South Central (Berkshire) NHS research ethics committee (NHS REC H0102/45). Samples were all obtained from the Histopathology Department of Southmead Hospital (Bristol, UK). Samples were collected in 10% neutral buffered formalin, embedded in paraffin and cut using standard techniques. The glycocalyx was labelled with UEA lectin and R18 as described previously[3]. All samples were disposed of after imaging was complete in accordance with local guidance.

### Placenta perfusion

Placentae were obtained from women with uncomplicated pregnancy undergoing elective cesarian section at term with full consent as part of the SPADE study. Qdot™ Streptavidin conjugates (Thermo Fisher Scientific, Waltham, USA, Q10123MP) were freshly prepared in 1% BSA PBS – 0.1% Tween, pH 6.8 with LEL lectin. Tissue was incubated at room temperature for 1 h. The sections were then post-fixed with 1% glutaraldehyde in 0.1 M phosphate buffer and processed and imaged using standard techniques on Tecnai 12 – FEI BioTwin Spirit (Field Electron and Ion Company, Hillsboro, USA) TEM. All samples were disposed of after imaging was complete in accordance with local guidance.

### RBC lectin staining

Washed, fixed blood smears were blocked using a 1% BSA (VWR, USA, 422361V) PBS solution for 30 min and washed 3 times before incubating with Alexa fluor conjugated lectins suspended in PBS at a concentration of 1:200 (LEL) (ThermoFisher Scientific, L32478) or WGA (GTX01502; GeneTex; 5 mg/ml; 1:500–1:1000) or MOA (Z8-BA-9001-1, TCS Biosciences, 2 mg/mL; 1:100) overnight at 4 degrees. After further washing 36.5 µg/ml Octadecyl rhodamine B chloride (R18) (previously suspended in 1 ml of 100% ethanol to make a stock solution) was further diluted 1:1000 in PBS immediately prior to each use (O246; ThermoFisher Scientific) and added to slides for 10 min. Slides were then washed a further 3 times in sterile PBS (pH 7.4) before Vectashield mounting medium (H-1000; Vector Laboratories) and a coverslip were applied.

### Imaging

RBC smears were imaged using either an AF600 LX wide-field fluorescence microscope (Leica Microsystems) (100× Oil immersion lens, NA 1.25, pixel size set to 60 nm) or a Leica SP8 confocal microscope (60x oil immersion lens, NA 1.4, pixel size set to 52 nm). For 'peak to peak' analysis suitably spaced RBC were selected 'blind' to the lectin signal before imaging 3 discrete areas of the smear while avoiding white blood cells. The image 'z' focal position was chosen for each image to ensure RBC were consistently imaged through their axis at their widest point as shown in Fig. 1c (a representative image of a healthy RBC from the SPADE study).

### RBC 'peak to peak' assessment

The RBC 'peak to peak' analysis method is adapted from existing validated, published techniques[3]. Where automated analysis was employed RBC within images were detected using a custom StarDist model[3,53]. Image J (FIJI) Intensity profiles perpendicular to object surfaces were measured using a custom workflow created using the ModularImageAnalysis (MIA) plugin for Image J[54–56]. At regular intervals along the surface of each detected object, intensity profiles were extracted perpendicular to the surface at that point. An asymmetric Gaussian profile was fit to each profile, where $a$ is the peak amplitude, $b$ is the peak maxima location, $c$ and $f$ are the peak standard deviations (widths) to the left and right of the maxima, $d$ and $e$ are the baselines (background) to the left and right of the maxima and $k$ controls the transition between left and right sides (set to 100 to effectively give an

immediate transition).

$$y = \left(1 - \frac{1}{1 + \exp(-2k(x - b))}\right)\left(d + (a - d)exp\left(-\frac{(x - b)^2}{2c^2}\right)\right)$$
$$+ \left(\frac{1}{1 + \exp(-2k(x - b))}\right)\left(e + (a - e)exp\left(\frac{-(x - b)^2}{2f^2}\right)\right)$$

This equation utilises the Heaviside step function to yield a profile with different fitting parameters to the left and right of the peak maxima but retain a common peak maxima location. Each parameter was subject to user-defined constraints and likewise, only fits with acceptable $R^2$ values were retained for analysis. By fitting such profiles to two different signals, it was possible to measure the relative difference in peak maxima location. Simultaneously the MIA plugin can generate images to highlight RBC selection (Fig. 1e) and RBC analysis (Fig. 1f) for each confocal image used.

### Glomerular permeability assay
The glomerular permeability assay records the rate of diffusion of labelled albumin inside glomeruli into the surrounding unlabelled albumin[3,23]. Ringer perfused rat kidneys were sieved in 4% BSA Ringer solution to isolate glomeruli. Isolated glomeruli were incubated in 36.5 µg/mL octadecyl rhodamine B chloride (R18) (O246; Thermo Fisher Scientific) for 15 min and were then washed in 4% Ringer BSA to remove unbound R18, followed by 15 min incubation in 30 µg/mL Alexa Fluor 488-BSA (A13100; Thermo Fisher Scientific). Individual glomeruli were trapped on a custom-made petri dish, and the perfusate was switched from 30 µg/mL labelled 488-BSA to 30 µg/mL unlabelled BSA. A Nikon Ti-E inverted confocal microscope (Nikon Instruments Inc.) was used to capture the fluorescence intensity. The rate of decline in fluorescence intensity within the capillary loops for the first minute was used to calculate Ps'alb as previously described[3]. Observers were blinded to sample identity.

### Enzyme depletion and RBC Glx recovery
To deplete the RBCGlx isolated, washed, RBC were incubated in the presence of neuraminidase (Roche, 11 585 886 001) or heparinase III (Sigma H8891). 100,000 RBC per ml of media were subsequently added to 10 cm round culture dishes containing endothelial monolayers. To induce interaction movement of the RBC in suspension over the static monolayers was induced with an orbital shaker resulting in a modelled peak shear stress of 5 dyn/cm$^2$ (SSM1 Stuart UK)[57].

### Generation and validation of EXT1 over expressing endothelial cells
Human glomerular endothelial cells[58] were transduced with lentivirus carrying human EXT1-Myc-DKK transgene. HS over expression was confirmed using immunofluorescence. Anti-HS antibody (BIO-RAD, cat no. 1698) was added to cells at 4 degrees for 12 h following fixation (in 4% paraformaldehyde) and blocking (5% bovine serum albumin). Cells were imaged blind using fixed confocal microscope settings or imaged and analysed using an IN Cell analyser 2200 (GE Healthcare) and cell analyser workstation software (GE Healthcare) (Supplementary Fig. 8)[59].

### GlycoCheck™
Sublingual images were captured using a CapiScope HVCS Handheld Capillary Microscope (KK Technology, Honiton, England). Videos of the sublingual microcirculation were acquired and analysed by the automated capture and analysis system GlycoCheck™ [version 2.0] (Microvascular Health Solutions Inc., Salt Lake City, Utah, USA) in line with previous published protocols.

### Endothelial culture
Human conditionally immortalised glomerular endothelial cells were grown and maintained in EGM2 media (Lonza) supplemented with 10% FBS and EGM2-MV bullet kit (Lonza) in the absence of gentamicin[58]. Cells were differentiated at 37 °C for 5 days (80% confluence) before entering the study and were free of mycoplasma infection. To mimic a diabetic environment in vitro, endothelial cells were maintained in the presence of 100 nmol/L insulin (Tocris), 25 mmol/L glucose (MilliporeSigma), 1 ng/mL TNF-α, and 1 ng/mL IL-6 (both from R&D Systems)[3].

### Click chemistry transfer experiments
To integrate ManNAz into the eGlx, terminally differentiated confluent endothelial monolayers (grown in EBM2 (Lonza, cc1356) as previously reported)[3] were exposed to EBM2 media supplemented with ManNAz (25 µg/ml or DMSO (solvent control) for 5 days (Thermo Scientific, C33366). To label cells the Click-iT™ cell reaction buffer kit was used (Thermo Scientific, C10269) to conjugate Alexa Fluor™ 488 Azide to the integrated sugar. The standard recommended protocol was followed throughout. For transfer experiments endothelial cells were washed 3 times with PBS before standard EBM2 was added containing a suspension of RBC for interaction. Retrieved RBC were washed 3 times (with centrifuge isolation cycles (200 rcf, 5 min)) before transfer to new unlabelled monolayers at a concentration of 100,000 RBC per ml of media. At the experimental end RBC were removed from the monolayers before washing the endothelial cells 3 times and fixing with PFA (and labelling the cells per the cell reaction buffer kit protocol). RBC were washed and pipetted onto glass slides for labelling.

### Reporting summary
Further information on research design is available in the Nature Portfolio Reporting Summary linked to this article.

## Data availability
The data supporting the findings from this study are available within the manuscript and its supplementary information. Any additional raw data is available from the corresponding author upon reasonable request. Source data are provided with this paper.

## Code availability
The automated analysis software is frequently updated to ensure ongoing compatibility with FIJI. The software will be made available from the corresponding author upon request.

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

## Acknowledgements

MRC fellowship MR/W024187/1, KRUK IN_004_20190305 (M.J.B.), Diabetes UK grant 18/0005795 (R.R.R.), KRUK grant KS_RO_002_20190917 (R.R.R.), BHF grant PG/22/11121 (R.R.R.), KRUK grant RP_031_20180306 (M.C.), BHF grant PG/20/7/34849 (Y.Q.), BHF clinical research training fellowship FS/CRTF/22/24361 (L.S.), National Institutes of Health grant R01 AR055670 (Y.Y.), The DISCOVER study was supported by grants from the Southmead Hospital Charity and Elizabeth Blackwell Institute. The SPADE study was supported by the David telling charitable trust. The authors gratefully acknowledge the Wolfson Bioimaging Facility for their support and assistance in this work.

## Author contributions

M.J.B. performed experimental studies, carried out analysis and secured funding for the studies. R.R.R. performed experimental studies, carried out analysis. M.C. performed experimental studies, carried out analysis. J.A. performed experimental studies, carried out analysis. M.G. performed experimental studies. C.D. performed experimental studies. C.H. performed experimental studies. C.N. performed experimental studies. J.L. performed experimental studies. Y.Q. performed experimental studies. L.C. performed experimental studies. L.S. performed experimental studies. S.C. developed analysis software. Y.Y. provided material resources. J.S. provided materials resources. V.B. provided materials resources. G.I.W. Supervised the work and helped to secure funding for the project. R.R.F. supervised the work and helped to secure funding for the project. S.C.S. supervised the work and helped to secure funding.

## Competing interests

A patent relating to the concepts outlined in this manuscript has been filed by the University of Bristol (application number GB2410240.2). M.J.B. and S.C.S. are named authors on the patent. M.J.B. and S.C.S. are cofounders of a university spin out company (Dimension Biotech) developing methods to routinely measure glycocalyx integrity. M.J.B. and S.C.S. own shares in this company. The remaining authors have no competing interests.
