## [Transparent Peer Review file · Nature Communications]

Endothelial-erythrocyte glycocalyx exchange enables liquid biopsies of endothelial function

Corresponding Author: Dr Matthew Butler

Version 0:

Reviewer comments:

Reviewer #1

(Remarks to the Author)

In this manuscript, Butler et al. provide data demonstrating exchange of endothelial glycocalyx (eGlx) with the glycocalyx of red blood cells (RBCGlx), presumably through physical interactions. To demonstrate this, the authors of the study employ a novel RBC analysis method, which combines confocal microscopy and artificial intelligence software. They analyze various model organisms, spanning from human tissue and blood samples, to rodents and in vitro cell cultures. The authors claim to be able to use the RBC glycocalyx analysis to estimate damage to the endothelial glycocalyx, thus moving towards more effective diagnostics procedures.

Overall, it is unclear whether this claim is demonstrated.

First, the importance of the method depends on its ability to distinguish more than just a general sickness state of the patient. However, very different diseases (from COVID19 infection to kidney disease) showed the same phenotype, a reduction of RBC glycocalyx, which raises an important question about the applicability of the method to real-world diagnostics.

Second, the authors claim that their discovery 'represents a leap in understanding of blood vessel function'. Nevertheless, they do not present data on the mechanistic aspect of the glycocalyx exchange between eGlx and RBCGlx. Therefore, the functional aspect of this process remains elusive.

Third, the confocal imaging method utilized in this study, relies on a complex distance measurement, requiring two-color labeling of RBC. It seems that this method is too complicated to be used for screening purposes. This reviewer is not convinced that this particular method provides any additional information to the simple investigation of the RBC glycocalyx intensity, which could be assessed by simple means, either through single-color fluorescence imaging, or even by non-fluorescence means (e.g. biochemistry), and would be more suitable for screening purposes.

Additional Major points.

4. The method in Fig. 1 appears novel. However, it has not been calibrated against a range of intensities. When the glycocalyx is dimmer, does it appear to be closer to the membrane? This could be tested by using different labels of FITC lectin.

5. It is not entirely clear how Fig 2A should be interpreted. It should be either properly explained, or it should be a supplement to the main manuscript.

6. For the experiments in Fig. 3A, it would be meaningful to include a control tissue, if possible. It is not clear why the authors measure distances in Fig 3E, when the difference in intensities is brutal, and is clearly sufficient for any differentiation between the phenotypes.

7. Fig. 4 appears to support the main claims of the authors. It could be moved earlier in the manuscript. The number of experiments in this figure is not clear and it should be introduced. Same as Fig4, Fig. 6 and 7 could be moved earlier in the manuscript

8. Fig. 5 should include representative images.

9. The manuscript is poorly written, with many awkward phrases, some of which are nonsensical. Was the manuscript written using automated AI processing? If so, this should be declared, and errors should be carefully corrected.

10. When hovering over different figures, unusual messages appear (see screenshots below), suggesting AI-based interpretation of the material. This again raises the concern that the work was largely based on AI writing, which was then insufficiently checked by the authors.

Minor points

In Fig. 1D numbers are missing on the x and y axis.

In Fig. 1E will benefit from arrows specifying what platelets and what white blood cells are.

The description of Fig. 1F in the figure legend is missing.

For better reading, the nature of the tissue in line 73 should be added. It is written in the figure description but not in the main text.

It is not clear if the RBC in Fig 3A are from the same field of view or not.

It is not clear what does the circles in Fig 3B and C represent. Are these number of patients, biopsies, cells?

Fig. 4A is not mentioned in the main text, but only in the figure description.

Line 165 should be 'as previously described'.

It is not clear why the word 'increased' in the description of Fig. 4 is underlined.

The units in Fig 7 E and H are missing, as well as, the scale bar in G.

In line 269, the word Fluorescent is spelled wrong – 'Florescent'. Also, the whole sentence does not make any sense.

It is not clear why 'bidirectional transfer' in line 302 is underlined.

The sentence 'Heparan sulphate proteoglycans, which are highly sulphated and tend to contain sialated glycans, a major component of the eGlx.' is missing a verb.

In line 384 'Data gathered to date suggests', data is plural, suggests should be corrected to suggest.

Line 445 (in ethanol was diluted 1:1000 in PBS) – unusual formulation.

The addition of objective's NA will strengthen the methods section.

Supplementary Fig. 1A It is not clear what do the circles represent. Also, the color of the circles is the same as the lines showing mean/median.

Screenshots

Reviewer #2

(Remarks to the Author)

Dr Butler and co-workers present a multifaceted study of interactions of glycocalyx of RBC and endothelial cells. The study is characterized by engagement of highly sophisticated complementary tools to not only yield the novel conclusions on interchange of glycocalyx of RBCs and endothelial cells in two rodent models and in humans, but has a translational aspect important for a non-invasive diagnostics ("liquid biopsy".) Overall, these are most exciting and innovative investigations into the workings of microcirculation.

I have a few questions/comments related to this enticing, thought-provoking work.

1. Wouldn't conclusions made using ManNAz click probe be further strengthened by demonstration of its disappearance following treatment with heparinase?
2. Considering the intensity of renewal of components of endothelial glycocalyx, wouldn't 4-24h period of contact between EC and RBC a time sufficient for the synthesis of at least some of them.
3. Fig 7,B appears to be out of focus.
4. Fig 4, B – treatment with neuraminidase increasing staining. Assuming that the explanation presented is correct and considering the repeated use of neuraminidase in the study to damage glycocalyx, wouldn't this pseudo-enhanced staining jeopardize other staining results?
5. One of the earliest studies of RBC-EC interactions by Oberleitner (referenced in this manuscript) described "fingerprinting" process that may propagate glycocalyx injury from one cell type to another. How can one reconcile this old study utilizing atomic force microscopy with the present finding of a mutual exchange of glycocalyx between these cell types?
6. In the same vein, previous studies utilizing mathematical modeling (Biophysical Journal 120, 1–12, August 3, 2021 and Matrix Biol Plus <https://doi.org/10.1016/j.mbplus.2021.100087>) predicted a "lifting" force acting on RBC from EC. Admittedly, this force wouldn't preclude periodic collision of two cell types. Perhaps, this consideration deserves a comment.
7. Finally, one could imagine scenarios for the glycocalyx exchange described in the manuscript:
 - A) Could it occur via microvesicles? Wouldn't a conditioned medium still contain them?
 - B) Is it possible that during the process of "fingerprinting" the exchange occurs while the damage to glycocalyx still sustains?

Michael S Goligorsky

Reviewer #3

(Remarks to the Author)

Dear editor-in-chief:

Thank you for your invitation to review. Authors introduces a novel methodology for glycocalyx (Glx) detection and demonstrates through in vitro and in vivo experiments that endothelial cells (ECs) and red blood cells (RBCs) exhibit bidirectional exchange of glycocalyx components under pathological conditions.

The study addresses an interesting scientific question, and several methodological concerns regarding the experimental validation require clarification:

Major Concerns:

1. Given the demonstrated bidirectional transfer phenomenon, could erythrocytes potentially transfer glycocalyx components to endothelial cells under EXT1-deficient conditions?

2. While the novel detection methodology appears more suitable for in vitro or ex vivo tissue section analyses compared to existing GlycoCheck™ technology, its application in the final study phase utilized an in vivo model. Does this imply the technique is exclusively applicable to ex vivo specimens?

3. How do the authors define the term "liquid biopsies" in the article title? If non-applicable to in-vivo models, does this methodology present more constrained clinical applicability for glycocalyx assessment compared to GlycoCheck™?

4. The employment of diverse disease models (viral pneumonia and autoimmune nephropathy) raises the question: Is this bidirectional Glx transfer between RBCGlx and eGlx universally present in all vasculopathic conditions? Furthermore, is there a positive correlation between this bidirectional transfer and vascular permeability?

Minor Concerns:

The resolution of key figures appears insufficient.

Version 1:

Reviewer comments:

Reviewer #1

(Remarks to the Author)

The authors replied convincingly to most of my comments. However, the most critical aspect of the work has not been addressed.

In my original comment, I stated: "the confocal imaging method utilized in this study, relies on a complex distance measurement, requiring two-color labeling of RBC. It seems that this method is too complicated to be used for screening purposes. This reviewer is not convinced that this particular method provides any additional information to the simple investigation of the RBC glycocalyx intensity, which could be assessed by simple means, either through single-color fluorescence imaging, or even by non-fluorescence means (e.g. biochemistry), and would be more suitable for screening purposes."

The authors reply as follows: "there are scientific reasons behind our decision to use imaging techniques and specifically the 'peak to peak' method on red blood cells. Our group has previously shown that 'peak to peak' provides the most accurate assessment of glycocalyx depth compared to the historical gold standard – electron microscopy (PMID 28524373). We have further shown that 'peak to peak' on endothelial cells predicts endothelial barrier function better than electron microscopy (PMID 36749631). Using the same measure on RBC makes direct comparisons possible. We find intensity-based measurements at the RBC surface to provide a good indication of glycocalyx damage when samples can be processed and imaged simultaneously. However, in clinical studies, blood samples are provided intermittently and from multiple sites. We have therefore developed a technique that can accurately assess glycocalyx changes over a range of lectin concentrations/laser intensities / lectin batches and have illustrated this robustness with new experimental data in supplemental figure 2. We do, however, agree that now we have shown the utility of RBCGlx monitoring, developing a simpler method will make its transition to clinical practice more likely and thank the reviewer for their suggestion."

This reply is not convincing. The two cited references do not deal with RBCs, and are therefore not relevant to this point. The argument that "blood samples are provided intermittently and from multiple sites" is also not convincing. In a clinical setting the blood samples would be treated with a standard lectin concentration, and would be imaged in parallel with a standard sample (e.g. fluorescent microspheres, of determined fluorophore concentration), so that intensity variations would be negligible and/or would be compensated for, by comparison with the standard sample.

To reply to this argument, the authors should simply test their method against a simpler one: either intensity measurements or the biochemical determination of glycocalyx components (for example by Western Blotting).

Reviewer #2

(Remarks to the Author)

I'm completely satisfied with authors rebuttal and additional studies performed in response to me suggestions.

Reviewer #3

(Remarks to the Author)

In the initial review, our primary concerns centered on two issues:

1. Whether the methodology is applicable for in vivo dynamic tracking;
2. Whether this glycocalyx exchange phenomenon occurs universally across all disorders associated with altered vascular permeability.

In the current response, the authors have addressed these queries with corresponding explanations. However, regarding the second issue, the evidence provided remains inconclusive and fails to substantiate the claim convincingly.

Furthermore, if this represents a ubiquitous inflammatory response, does detecting such exchange possess distinct clinical utility? Considering that vascular hyperpermeability manifests in virtually all inflammatory conditions, we recommend the authors incorporate additional experimental evidence or mechanistic explanations to validate the pathophysiological significance of this phenomenon.

Version 2:

Reviewer comments:

Reviewer #1

(Remarks to the Author)

The authors have now replies to my comment.

The reply is not perfect, as no statistical comparison between the two methods from Supplementary Fig. 5 is offered.

Nonetheless, the authors have, at least, made an effort to address this important issue, enabling readers to make their own evaluations of the tools provided by the authors. I am in agreement with the publication of the manuscript.

Reviewer #3

(Remarks to the Author)

I have read the authors' rebuttal and the revised manuscript. I am pleased with these changes. My main concerns about the clinical relevance of the test have been successfully addressed, and the authors have now provided excellent explanations.

Endothelial-erythrocyte glycocalyx exchange opens the door for ‘liquid biopsies’ of endothelial function

NCOMMS-25-16402A.

Reviewer 1.

“In this manuscript, Butler et al. provide data demonstrating exchange of endothelial glycocalyx (eGlx) with the glycocalyx of red blood cells (RBCGlx), presumably through physical interactions. To demonstrate this, the authors of the study employ a novel RBC analysis method, which combines confocal microscopy and artificial intelligence software. They analyze various model organisms, spanning from human tissue and blood samples, to rodents and in vitro cell cultures. The authors claim to be able to use the RBC glycocalyx analysis to estimate damage to the endothelial glycocalyx, thus moving towards more effective diagnostics procedures.

Overall, it is unclear whether this claim is demonstrated.

First, the importance of the method depends on its ability to distinguish more than just a general sickness state of the patient. However, very different diseases (from COVID19 infection to kidney disease) showed the same phenotype, a reduction of RBC glycocalyx, which raises an important question about the applicability of the method to real-world diagnostics.”

Many thanks to reviewer 1 for the detailed review. We are grateful for their positive comments on the novelty of the techniques we have developed during this work.

Addressing their first point - we have used a diverse selection of pathological insults to test our assay. Endothelial glycocalyx injury has been implicated in the early pathogenesis of many conditions. It was important, therefore, to show whether the identified relationship between red blood cells and endothelial cells was an isolated disease-specific observation, or more generally applicable. Our findings to date suggest glycocalyx injury can be detected on the red blood cell surface in multiple diseases once endothelial glycocalyx injury has occurred. Of course, this means that detecting red blood cell glycocalyx damage is not diagnostic for a single condition, but we do not feel that makes it any less useful. In clinical practice multiple tests are not diagnostic of a single condition. For example C-reactive protein (CRP) is used to monitor inflammatory responses and needs interpreting in context for each patient. We feel the same is true for glycocalyx assessments. Detecting early endothelial damage could be highly beneficial in acute care settings, including sepsis. In chronic health condition such as diabetes (where an individual’s risk of developing microvascular damage cannot entirely be explained by their HbA1C level) detecting glycocalyx injury may help to risk stratify patients. We are currently testing applications in both of these indications. We hope that our manuscript will further stimulate clinical glycocalyx research into other applications by providing an entirely new way to monitor glycocalyx injury with significant advantages over current methods.

“Second, the authors claim that their discovery ‘represents a leap in understanding of blood vessel function’. Nevertheless, they do not present data on the mechanistic

aspect of the glycocalyx exchange between eGlx and RBCGlx. Therefore, the functional aspect of this process remains elusive.”

Regarding the second point - we see the endothelial cells' ability to repair glycocalyx damage on the red blood cell surface as an entirely new functional role for endothelial cells however we have reworded the highlighted sentence 'represents a leap in understanding of blood vessel function' in the abstract (lines 24-26) to be more specific. It now states "These discoveries open the door to real-time monitoring of endothelial damage in patients whilst simultaneously providing a potential explanation as to how red blood cells maintain a glycocalyx during circulation despite limited synthetic capacity.”

“Third, the confocal imaging method utilized in this study, relies on a complex distance measurement, requiring two-color labeling of RBC. It seems that this method is too complicated to be used for screening purposes. This reviewer is not convinced that this particular method provides any additional information to the simple investigation of the RBC glycocalyx intensity, which could be assessed by simple means, either through single-color fluorescence imaging, or even by non-fluorescence means (e.g. biochemistry), and would be more suitable for screening purposes.

Additional Major points.

4. The method in Fig. 1 appears novel. However, it has not been calibrated against a range of intensities. When the glycocalyx is dimmer, does it appear to be closer to the membrane? This could be tested by using different labels of FITC lectin.”

Addressing points 3/4, as briefly outlined above, there are scientific reasons behind our decision to use imaging techniques and specifically the 'peak to peak' method on red blood cells. Our group has previously shown that 'peak to peak' provides the most accurate assessment of glycocalyx depth compared to the historical gold standard – electron microscopy (PMID 28524373). We have further shown that 'peak to peak' on endothelial cells predicts endothelial barrier function better than electron microscopy (PMID 36749631). Using the same measure on RBC makes direct comparisons possible. We find intensity-based measurements at the RBC surface to provide a good indication of glycocalyx damage when samples can be processed and imaged simultaneously. However, in clinical studies, blood samples are provided intermittently and from multiple sites. We have therefore developed a technique that can accurately assess glycocalyx changes over a range of lectin concentrations/ laser intensities / lectin batches and have illustrated this robustness with new experimental data in supplemental figure 2. We do, however, agree that now we have shown the utility of RBCGlx monitoring, developing a simpler method will make its transition to clinical practice more likely and thank the reviewer for their suggestion.

“5. It is not entirely clear how Fig 2A should be interpreted. It should be either properly explained, or it should be a supplement to the main manuscript.”

Point 5 – we have moved this additional figure to the supplement (supplemental figure 3) and added additional text to explain the protocol used more fully highlighting how the model of diabetes is established, when treatment was started and when blood samples were taken.

“6. For the experiments in Fig. 3A, it would be meaningful to include a control tissue, if possible. It is not clear why the authors measure distances in Fig 3E, when the difference in intensities is brutal, and is clearly sufficient for any differentiation between the phenotypes.”

Point 6- we have added a more detailed explanation about why we included thin basement membrane disease (TMD) samples as reference samples (page 3 lines 18-21). The method of sample collection, fixation and storage all influence glycocalyx preservation and so we believe these TMD samples are the best comparison group currently available. This is because they are taken and processed identically to the minimal change nephrotic syndrome (MCNS) samples.¹

“7. Fig. 4 appears to support the main claims of the authors. It could be moved earlier in the manuscript. The number of experiments in this figure is not clear and it should be introduced. Same as Fig4, Fig. 6 and 7 could be moved earlier in the manuscript”

Point 7 – we are pleased the reviewer agreed that figures 4,6 and 7 support the hypothesis and we have followed their suggestions in updating the figure legend to include the number of experimental repeats. We believe the broad reader base of Nature Communications will be most interested in the clinical applications of the developed technology and the human data. For this reason, we have included these figures before those investigating the mechanism. However, we have added additional text to link the sections better throughout the manuscript. We hope these additions improve the manuscript flow and make the figure order more logical (page 2 lines 39-40, page 3 line 11, page 4 lines 6-8 and 31-33).

“8. Fig. 5 should include representative images.”

Point 8 – We have added additional representative images to figure 5.

“9. The manuscript is poorly written, with many awkward phrases, some of which are nonsensical. Was the manuscript written using automated AI processing? If so, this should be declared, and errors should be carefully corrected.

10. When hovering over different figures, unusual messages appear (see screenshots below), suggesting AI-based interpretation of the material. This again raises the concern that the work was largely based on AI writing, which was then insufficiently checked by the authors.”

Points 9/10 – We have spent further time proof-reading the manuscript and revising it to improve the sentence structure and flow. Most of the authors are native English speakers, but we apologise if some sentences explaining the complex

methodology/results appeared awkward. AI has not been used at any stage in the preparation or proof reading of the manuscript. In addition, AI was not used in the preparation of any of the figures. We cannot explain why unusual messages appeared when hovering over the figures as they were not present on the version we submitted, and we have checked the current version and can't see any such messages.

Minor points

"In Fig. 1D numbers are missing on the x and y axis."

Fig 1D was included to provide an indication of the 'shape' of the light intensity profiles obtained, but we have now added numeric axis labels as requested.

"In Fig. 1E will benefit from arrows specifying what platelets and what white blood cells are. The description of Fig. 1F in the figure legend is missing."

Figure 1E has been updated to highlight the platelets (which are not included in the analysis by the software). There are no white blood cells in the images presented. Areas of the blood smear including white blood cells are not routinely imaged and so we have updated the figure legend to reflect this. It now states "Artificial Intelligence was trained to identify RBC on blood smears whilst ignoring adjacent platelets."

Figure 1F – the legend has been updated and now states "Identified RBC were examined automatically with line profiles measured every 15 degrees around the circumference of each RBC, white lines indicate included profiles, blue lines indicate automatically excluded measures. Profiles are excluded where the light profiles are interrupted by the signal from adjacent cells"

"For better reading, the nature of the tissue in line 73 should be added. It is written in the figure description but not in the main text."

Line 73 has been updated to highlight that a segment of glomerulus is illustrated, now shown on (Page 2 line 2).

"It is not clear if the RBC in Fig 3A are from the same field of view or not. It is not clear what does the circles in Fig 3B and C represent. Are these number of patients, biopsies, cells?"

Figure 3A – further details on the inset image have now been added and the legend for figures 3B and C updated. The figure legend now includes "Inset images display the green channel only on the indicated sections at higher magnification to highlight RBCGlx variations"

"Fig. 4A is not mentioned in the main text, but only in the figure description. Line 165 should be 'as previously described'. It is not clear why the word 'increased' in the description of Fig. 4 is underlined. The units in Fig 7 E and H are missing, as well as, the scale bar in G. In line 269, the word Fluorescent is spelled wrong – 'Florescent'.

Also, the whole sentence does not make any sense. It is not clear why 'bidirectional transfer' in line 302 is underlined. The sentence 'Heparan sulphate is a known binding site for malarial parasites on the RBC surface.² Heparan sulphate proteoglycans, which are highly sulphated and tend to contain sialated glycans, a major component of the eGlx.³ sulphate proteoglycans, which are highly sulphated and tend to contain sialated glycans, a major component of the eGlx.' is missing a verb. In line 384 'Data gathered to date suggests', data is plural, suggests should be corrected to suggest. Line 445 (in ethanol was diluted 1:1000 in PBS) – unusual formulation. The addition of objective's NA will strengthen the methods section.

Supplementary Fig. 1A It is not clear what do the circles represent. Also, the color of the circles is the same as the lines showing mean/median.

Screenshots”

Many thanks for these detailed observations. We have adjusted the manuscript to address each of these points as follows:

- Figure 4A is now referenced more clearly in the text (page 4 line 13)
- line 165 has been updated (page 3 line 13).
- The units and scale bars have been updated in figures 7 E,H and G.
- The miss spelt 'Fluorescent' has been corrected (page 5 line 23) and the sentence now reads “Lectins with fluorescent tags rapidly transferred onto the endothelial surface and the transfer appeared to be increased when monolayers were enzymatically damaged before interaction (supplemental figure 9).”
- The underlined section has been corrected (page 6 line 4). The sentence 'Heparan sulphate proteoglycans, which are highly sulphated and tend to contain sialated glycans, a major component of the eGlx.' Has been changed to “HS is a highly sulphated glycosaminoglycan and a major component of the eGlx.³ HS is also a known binding site for malarial parasites on the RBC surface.²”
- Line 384 has been corrected to “Data gathered to date suggest” (page 7 line 42)
- The R18 dilution has been clarified “Octadecyl rhodamine B chloride (R18) suspended in 1 ml of 100% ethanol to make a stock solution was further diluted 1:1000 in PBS immediately prior to each use” (page 8 line 49-51)
- We have added microscope objectives (page 9 lines 6 and 7).
- The legend associated with image in supplemental figure 1A now reads “A 3D reconstruction generated from a z-stack of confocal images of RBCs labelled with LEL lectin (green) and R18 (red).”
- The remaining figures are now submitted in the requested format for Nature communications publication.

Reviewer 2.

“Dr Butler and co-workers present a multifaceted study of interactions of glycocalyx of RBC and endothelial cells. The study is characterized by engagement of highly sophisticated complementary tools to not only yield the novel conclusions on interchange of glycocalyx of RBCs and endothelial cells in two rodent models and in humans, but has a translational aspect important for a non-invasive diagnostics (“liquid

biopsy”). Overall, these are most exciting and innovative investigations into the workings of microcirculation.

I have a few questions/comments related to this enticing, thought-provoking work.”

We would like to thank reviewer 2 for highlighting the highly sophisticated complimentary tools we have used to test our novel hypothesis. We are pleased you agree that this work is exciting and innovative.

“1. Wouldn’t conclusions made using ManNAz click probe be further strengthened by demonstration of its disappearance following treatment with heparinase?”

We have now added additional experimental work confirming that the MaNAz label on endothelial cells is located within the glycocalyx and is depleted by glycocalyx cleaving enzymes. We discuss this addition on page 5 lines line 32 and 33 and include new images and quantification data in supplemental figure 10.

Considering the intensity of renewal of components of endothelial glycocalyx, wouldn’t 4-24h period of contact between EC and RBC a time sufficient for the synthesis of at least some of them.”

Point 2- we agree endothelial cells can rapidly repair their glycocalyx following damage. For this reason, lectin studies were limited to studying transfer onto the red blood cells (which have limited regenerative capacity). Using ‘Click’ labelled sialic acids to study transfer onto endothelial cells ensured that transfer could be studied specifically as the two cell types swap glycocalyx. As a result, we did not have to remove native glycocalyx on endothelial cells to study this transfer and so could avoid any effects due to regenerating endothelial glycocalyx.

“3. Fig 7,B appears to be out of focus.”

Point 3. We have now included a new image.

“4. Fig 4, B – treatment with neuraminidase increasing staining. Assuming that the explanation presented is correct and considering the repeated use of neuraminidase in the study to damage glycocalyx, wouldn’t this pseudo-enhanced staining jeopardize other staining results?”

Point 4. This is an interesting point. We have found that peanut lectin (PNA) and wisteria lectin (WFL) binding increases at the cell surface following neuraminidase exposure as published previously. Simultaneously wheat germ agglutinin (WGA) binding reduces. Each lectin is used in isolation on each sample. We have not used PNA or WFL for ‘peak to peak’ measures because the absence of binding (PNA) or limited binding (WFA) on control RBCGlx makes them poorly suited to this technique.

However, the ‘peak to peak’ measure of glycocalyx remains valid across a wide range of intensities as shown by our recent study (supplemental figure 2) and so we would not

expect isolated alterations in intensity to affect the 'peak to peak' (depth) measure. This is illustrated following RBC exposure to the lowest concentration of neuraminidase used in supplemental figure 1 (1mU/ml for 30 minutes). After exposure we see marked reductions in the intensity of LEL staining within the RBCGlx but relatively preserved RBCGlx depth in most cells. These data further suggest that the intensity of lectin staining and 'peak to peak' measure of depth provide 2 separate measures of RBCGlx changes.

"5. One of the earliest studies of RBC-EC interactions by Oberleitner (referenced in this manuscript) described "fingerprinting" process that may propagate glycocalyx injury from one cell type to another. How can one reconcile this old study utilizing atomic force microscopy with the present finding of a mutual exchange of glycocalyx between these cell types?"

Point 5. Oberleithner's in vitro work suggested that interacting red blood cells with a damaged endothelium resulted in loss of glycocalyx from the red blood cell surface after 5 hours of interaction (measured using atomic force microscopy). However, he did not show where the red blood cell glycocalyx went. From the paper, it is not possible to see if the glycocalyx on enzyme-exposed cells was repaired following interaction. We propose that following interaction, the red blood cells 'donated' glycocalyx components to the damaged endothelial monolayer. We have added an additional line of discussion to highlight this interesting interpretation of Oberleithner's historical work (page 6 lines 48-51). However, flow induced by a rocker inverting 4 times a minute (as used in Oberleithner's study) will be markedly different from both the laminar flow we have modelled and the flow patterns seen in vivo and these differences make direct comparison of the data impossible.

"6. In the same vein, previous studies utilizing mathematical modeling (Biophysical Journal 120, 1–12, August 3, 2021 and Matrix Biol Plus <https://doi.org/10.1016/j.mbplus.2021.100087>) predicted a "lifting" force acting on RBC from EC. Admittedly, this force wouldn't preclude periodic collision of two cell types. Perhaps, this consideration deserves a comment."

Point 6. We have now highlighted recent work in this field in our discussion (page 7 lines 9-11). As noted, occasional collisions are expected even under healthy conditions, but the repulsive force between cells is dependent on two intact glycocalyx structures, the absolute dimensions of which are still unknown. We hope, by highlighting the potential importance of these contact events, more research will be stimulated in this area (page 7 line 16-17).

"7. Finally, one could imagine scenarios for the glycocalyx exchange described in the manuscript:

- A) Could it occur via microvesicles? Wouldn't a conditioned medium still contain them?
- B) Is it possible that during the process of "fingerprinting" the exchange occurs while the damage to glycocalyx still sustains?"

Point 7a. Micro vesicle exchange was a leading hypothesis when we first noted the mirroring effect seen between the cell types. This was the primary reason for including conditioned media controls in the transfer experiments. However, we consistently saw no effect on RBCGlx recovery in the absence of direct contact.

Point b. We do not yet know if the exchanged glycocalyx components form a fully functional glycocalyx. As you note, endothelial cells regenerate their glycocalyx, which makes the isolated study of transferred glycocalyx function on the endothelial surface impossible. However, we have shown that the RBCGlx depth can be fully restored through endothelial interaction. We thank the reviewer for the suggestion to look at the function of this layer and we will plan this future work.

Reviewer 3.

“Thank you for your invitation to review. Authors introduces a novel methodology for glycocalyx (Glx) detection and demonstrates through in vitro and in vivo experiments that endothelial cells (ECs) and red blood cells (RBCs) exhibit bidirectional exchange of glycocalyx components under pathological conditions. “

Again, we would like to thank the reviewer for noting the novel methodology and interesting scientific question addressed.

Addressing specific points

“1. Given the demonstrated bidirectional transfer phenomenon, could erythrocytes potentially transfer glycocalyx components to endothelial cells under EXT1-deficient conditions?”

We have shown that sialic acids, which decorate terminal residues of many carbohydrates, transfer between endothelial cells and RBC. Based on our work it would be possible for healthy RBC with an intact RBCGlx to transfer glycocalyx components to EXT 1 deficient endothelial cells. However, in our studies RBC from these mice had a reduced RBCGlx. The endothelial EXT 1 knock-down mice also had significantly reduced eGlx glycocalyx compared to littermate controls.³ These data suggest that RBC derived glycocalyx is insufficient to restore the damaged eGlx in this model.

“2. While the novel detection methodology appears more suitable for in vitro or ex vivo tissue section analyses compared to existing GlycoCheck™ technology, its application in the final study phase utilized an in vivo model. Does this imply the technique is exclusively applicable to ex vivo specimens?”

3. How do the authors define the term "liquid biopsies" in the article title? If non-applicable to in-vivo models, does this methodology present more constrained clinical applicability for glycocalyx assessment compared to GlycoCheck™?"

Our detection methodology is highly suitable for use on peripheral blood samples taken from in vivo experimental models or human study participants (as we have demonstrated in the examples described). We are unsure if the reviewer is asking about the utility of the developed 'peak to peak' assay on ex vivo blood samples or whether the red blood cell glycocalyx could be studied in vivo (non-invasively)? We have presented extensive data in the paper where we have used peripherally sampled venous blood to measure glycocalyx changes and shown they predict direct assessments of endothelial glycocalyx integrity. In addition, we have shown that red blood cell glycocalyx changes correlate with sublingual GlycoCheck™ measurements in human trials. We find analysing a simple blood sample to have significant advantages over GlycoCheck™, including patient acceptability, reduced variability and reduced cost. We do not feel analysing red blood cell glycocalyx in vivo (in situ) is necessary given the availability of blood samples in clinical practice and the small volume of blood needed for assessment (3 microlitres).

"4. The employment of diverse disease models (viral pneumonia and autoimmune nephropathy) raises the question: Is this bidirectional Glx transfer between RBCGlx and eGlx universally present in all vasculopathic conditions? Furthermore, is there a positive correlation between this bidirectional transfer and vascular permeability?"

Yes, data gathered to date suggests that bidirectional transfer occurs continually in health and disease. We believe this explains the correlations seen with GlycoCheck™ assessments in healthy pregnant women and with direct endothelial glycocalyx assessments in animal disease models. Glycocalyx injury is seen in a diverse array of pathologies and detecting this damage on red blood cells has provided an excellent marker of microvascular injury. We have presented data in figure 2 highlighting the strong negative correlation between the red blood cell glycocalyx depth on peripheral blood samples and the directly measured permeability of glomeruli to albumin. These data are expected given the multiple publications linking glycocalyx integrity and vascular permeability.

"Minor Concerns:

The resolution of key figures appears insufficient."

We have now ensured all images are submitted at a higher resolution.

1. Crompton, M., *et al.* Mineralocorticoid receptor antagonism in diabetes reduces albuminuria by preserving the glomerular endothelial glycocalyx. *JCI Insight* **8**(2023).
2. Vogt, A.M., Winter, G., Wahlgren, M. & Spillmann, D. Heparan sulphate identified on human erythrocytes: a Plasmodium falciparum receptor. *Biochem J* **381**, 593-597 (2004).

3. Gamez, M., *et al.* Heparanase inhibition as a systemic approach to protect the endothelial glycocalyx and prevent microvascular complications in diabetes. *Cardiovasc Diabetol* **23**, 50 (2024).

Endothelial-erythrocyte glycocalyx exchange opens the door for ‘liquid biopsies’ of endothelial function

Response to reviewers

We thank the reviewers for their ongoing review of our manuscript. We were pleased to see that reviewer 2 has no further concerns and that we have managed to answer the majority of reviewer 1 and 3's previous questions. We have highlighted the further alterations to the manuscript in yellow for ease.

Reviewer 1

“The authors replied convincingly to most of my comments. However, the most critical aspect of the work has not been addressed. In my original comment, I stated: “the confocal imaging method utilized in this study, relies on a complex distance measurement, requiring two-color labelling of RBC. It seems that this method is too complicated to be used for screening purposes. This reviewer is not convinced that this particular method provides any additional information to the simple investigation of the RBC glycocalyx intensity, which could be assessed by simple means, either through single-color fluorescence imaging, or even by non-fluorescence means (e.g. biochemistry), and would be more suitable for screening purposes.”

This reply is not convincing. The two cited references do not deal with RBCs, and are therefore not relevant to this point. The argument that “blood samples are provided intermittently and from multiple sites” is also not convincing. In a clinical setting the blood samples would be treated with a standard lectin concentration, and would be imaged in parallel with a standard sample (e.g. fluorescent microspheres, of determined fluorophore concentration), so that intensity variations would be negligible and/or would be compensated for, by comparison with the standard sample.

To reply to this argument, the authors should simply test their method against a simpler one: either intensity measurements or the biochemical determination of glycocalyx components (for example by Western Blotting).”

We are pleased that we have addressed the majority of reviewer 1's concerns and we thank the reviewer for their further suggestion.

In response to their new comments, we have studied RBC LEL surface intensity on samples taken from participants in the SPADE study. We have compared the intensity-based data and the ‘peak to peak’ based data to the results from side stream dark field imaging (GlycoCheck™ - PBR) in the same participants. These data are now presented in supplemental figure 5 and discussed on page 4 (lines 43-48) and page 11 (lines 20-31). Peak to peak measures correlate with the PBR value more strongly than intensity-based measures currently but we will continue to develop both these measures for future clinical studies.

Reviewer #2 (Remarks to the Author):

I'm completely satisfied with authors rebuttal and additional studies performed in response to me suggestions.

We thank reviewer 2 for their time reviewing our manuscript and we are pleased we have provided satisfactory responses to their questions.

Reviewer #3 (Remarks to the Author):

1. Whether the methodology is applicable for in vivo dynamic tracking;
2. Whether this glycocalyx exchange phenomenon occurs universally across all disorders associated with altered vascular permeability.

In the current response, the authors have addressed these queries with corresponding explanations. However, regarding the second issue, the evidence provided remains inconclusive and fails to substantiate the claim convincingly.

Furthermore, if this represents a ubiquitous inflammatory response, does detecting such exchange possess distinct clinical utility? Considering that vascular hyperpermeability manifests in virtually all inflammatory conditions, we recommend the authors incorporate additional experimental evidence or mechanistic explanations to validate the pathophysiological significance of this phenomenon.

Again, we thank the reviewer for their comments. We are pleased that we have answered point 1 and the other comments in the initial review.

Regarding the reviewer's second point "Whether this glycocalyx exchange phenomenon occurs universally across all disorders associated with altered vascular permeability".

In response to reviewer 3's questions we have confirmed that glycocalyx exchange continues to occur in an example disease model. Endothelial cells cultured under diabetic conditions continue to exchange glycocalyx components with circulating red blood cells. We have now included these data in supplemental figure 11 and discussed these data on page 10 (line 4-5) and page 12 (line 4-8).

The evidence we have gathered to date continues to suggest that glycocalyx exchange between endothelial cells and RBC is a physiological process. We believe the ongoing exchange, and the resulting dynamic equilibrium that forms between the two glycocalyx structures, remains relevant in pathological animal models and human disease. In vitro we have shown that this mechanism contributes to the close correlations we see between the RBC glycocalyx and the endothelial glycocalyx and explains why manipulating a single glycocalyx component within the endothelial glycocalyx results in measurable changes on the RBC (figures 5 and 6).

Regarding "mechanistic explanations to validate the pathophysiological significance of this phenomenon." Endothelial glycocalyx damage is an established cause of increased vascular permeability, a topic we reviewed in 2020 in the American Journal of Pathology (PMID: 32035881) and a topic we have been invited to review in more detail for a Nature Reviews Nephrology article which has been accepted for publication (reference number: NRNEPH-24-032V1D). This review will further highlight the importance of the endothelial glycocalyx as a permeability barrier and how its breakdown contributes to disease development making this manuscript highly topical.

The data presented in this manuscript suggest that the RBCGlx depth predicted glomerular albumin permeability in our diabetic rat model because it reflected changes occurring simultaneously on the fenestrated glomerular endothelial cells. The predictive power of the RBCGlx depth with regards to permeability will, however, depend on the pathology underlying the increased permeability (is glycocalyx loss the cause) and the site tested (following removal of the glycocalyx does a tight barrier remain). The ability of the RBCGlx to predict permeability changes therefore needs methodical targeted exploration before generalised assumptions are made.

Furthermore, if this represents a ubiquitous inflammatory response, does detecting such exchange possess distinct clinical utility?

The RBC peak to peak measurement does not detect exchange. It provides a measure of the RBCGlx depth which we have shown mirrors changes occurring simultaneously on the endothelial surface. The key question is - can measuring glycocalyx loss have clinical utility when it occurs in multiple pathological states? Published evidence based on less direct measures of glycocalyx damage suggest that it can.^{1, 2} As a practicing renal physician I use multiple tests that are non-specific for an individual pathology e.g. CRP, oxygen saturation, blood pressure and find they have significant utility. Detecting glycocalyx damage suggests that the endothelium is impaired and so it could provide a means of monitoring disease progression or treatment response. Test results should always be interpreted in clinical context; clinical judgement will help to decide if detected glycocalyx damage is due to diabetes or sepsis for example. Ultimately, however, clinical utility will be established through trialling glycocalyx monitoring in multiple clinical studies. We believe this is now an achievable goal having discovered a way of directly assessing glycocalyx depth changes on readily available clinical samples.

1. Uchimido R, Schmidt EP, Shapiro NI. The glycocalyx: a novel diagnostic and therapeutic target in sepsis. *Crit Care* **23**, 16 (2019).
2. Rizzo AN, Schmidt EP. The role of the alveolar epithelial glycocalyx in acute respiratory distress syndrome. *Am J Physiol Cell Physiol* **324**, C799-C806 (2023).